# Deep MALDI-MS spatial omics guided by quantum cascade laser mid-infrared imaging microscopy

Lars Gruber [1,2,9], Stefan Schmidt [1,9], Thomas Enzlein [1], Huong Giang Vo [3], Tobias Bausbacher [1,4], James Lucas Cairns [1,2], Yasemin Ucal[1], Florian Keller[1], Martina Kerndl[5,6], Denis Abu Sammour [1], Omar Sharif [5,7], Gernot Schabbauer [5,6], Rüdiger Rudolf [1,4], Matthias Eckhardt [8], Stefania Alexandra Iakab [1], Laura Bindila [3] & Carsten Hopf [1,2,4] ✉

In spatial'omics, highly confident molecular identifications are indispensable for the investigation of complex biology and for spatial biomarker discovery. However, current mass spectrometry imaging (MSI)-based spatial 'omics must compromise between data acquisition speed and biochemical profiling depth. Here, we introduce fast, label-free quantum cascade laser mid-infrared imaging microscopy (QCL-MIR imaging) to guide MSI to high-interest tissue regions as small as kidney glomeruli, cultured multicellular spheroid cores or single motor neurons. Focusing on smaller tissue areas enables extensive spatial lipid identifications by on-tissue tandem-MS employing imaging parallel reaction monitoring-Parallel Accumulation-Serial Fragmentation (iprm-PASEF). QCL-MIR imaging-guided MSI allowed for unequivocal on-tissue elucidation of 157 sulfatides selectively accumulating in kidneys of arylsulfatase A-deficient mice used as ground truth concept and provided chemical rationales for improvements to ion mobility prediction algorithms. Using this workflow, we characterized sclerotic spinal cord lesions in mice with experimental autoimmune encephalomyelitis (EAE), a model of multiple sclerosis, and identified upregulation of inflammation-related ceramide-1-phosphate and ceramide phosphatidylethanolamine as markers of white matter lipid remodeling. Taken together, widely applicable and fast QCL-MIR imaging-based guidance of MSI ensures that more time is available for exploration and validation of new biology by default on-tissue tandem-MS analysis.

Matrix-assisted laser desorption/ionization (MALDI) mass spectrometry imaging (MSI) is a fundamental label-free technology in spatial biology. It enables spatially resolved visualization, investigation and probabilistic mapping of lipids, metabolites, peptides, drugs or N-glycans in tissue sections in biomedical science as well as in clinical and pharmaceutical research[1–4]. Integration of whole tissue datasets sequentially acquired from the same or adjacent tissue sections by MSI

and orthogonal technologies such as (mid-)infrared (MIR) imaging, Raman imaging or spatial transcriptomics, often referred to as correlative spatial multi-omics[5–7], have opened up new avenues for scientific inquiry. Recent high-end MSI platforms, such as Fourier Transform MS (FTMS) or trapped ion mobility spectrometry (TIMS) MSI, offer superior **s**peed, **s**ensitivity, **s**patial resolution, or molecular **s**pecificity[8,9]. However, the four criteria in this 4S-paradigm of MSI

performance are currently mutually exclusive, and very high molecular specificity and sensitivity can only be obtained by in-depth spatial chemical analysis at lower speed and at reduced image resolution, i.e., large pixel sizes[10]. Therefore, in practice, most MALDI-MSI studies with high-performance instruments today appear to use slow FTMS imaging or timsTOF-MSI in TIMS-off/qTOF mode without using collisional cross section (CCS) information. Because of time constraints and despite the lack of HPLC separation in MSI, analytical capabilities for spatially resolved accurate mass determination (in FTMS imaging), on-tissue fragmentation analysis, and ion mobility separation of isobaric compounds (both in TIMS-MSI) are often not used. Without this multi-dimensional information, lipids/metabolites cannot be considered confidently identified.

To overcome instrument limitations, "smart" data processing and "smart" sampling methods (reviewed in ref. 11) such as histology- or MIR imaging-guided MSI have been suggested[12,13]. In the latter case, a non-destructive, label-free and fast "guiding" imaging modality determines the composition of tissue specimens by mid-infrared vibrational spectroscopy. It captures collective molecular information as a rough molecular sketch, i.e., relative lipid-, nucleic acid-, carbohydrate- and protein content, and allows for computational segmentation of the imaging data. This enables effective MIR-based definition of regions of interest (ROIs), e.g., the cerebellar granular layer in mouse brain[12]. ROI information is then transferred to the slower imaging mass spectrometer for focused MSI analysis of well-defined, often small tissue areas. This saves time and data volume, which could, in principle, be spent for advanced chemical bioanalysis with LC-MS-like analytical depth, i.e., with ultra-high resolving power provided by FTMS and/or by default use of TIMS in MALDI-MSI studies. Here, advances of recently described prototypical on-tissue parallel reaction monitoring-parallel accumulation and serial fragmentation (prm-PASEF) in LDI-MSI of tattoo pigments[14] and spatial ion mobility-scheduled exhaustive fragmentation (SIMSEF), a data-dependent acquisition technology that provides TIMS-MS imaging datasets with $MS^2$ spectra[15] may pave the way for ROI-focused in-depth spatial lipidomics with routine on-tissue $MS^2$.

However, MIR imaging-guided MSI has remained a mere concept so far, as available Fourier transform (FT-IR) imaging instruments are not fast enough for spatially focused tissue analysis at cellular resolution. In contrast, quantum cascade laser (QCL)-based MIR imaging microscopes feature a tunable coherent light source with high power density for high sensitivity and higher sample throughput in biological systems[16–20], thus enabling multiple new technologies and applications in MIR imaging utilizing vibrational probes[19,21], large scale plasmonic metasurfaces[22] or MIR-based whole slide scanning[18]. QCL-based MIR microscopes permit the selective acquisition of MIR data for user-defined single wavenumbers or full hyperspectral data. However, validated methods and dedicated computational tools for information-rich and high-throughput QCL-MIR imaging-guided MSI are lacking[7,17,23–25].

In addition to these challenges, method development and validation in MSI have generally been hampered by the lack of reliable analytical ground truths for segmentation and molecular identities[26,27], as the spatial and molecular composition of investigated tissues is typically unknown. To this end, synthetic datasets[2], expert crowdsourcing[26], single-cell fluorescence[28], or histopathology annotations[29] have been proposed as ground truths. To address this key challenge in MSI method development and validation, we propose that genetic mouse models with defined alterations in metabolism can be used as a qualitative ground truth. Leveraging QCL-MIR imaging-guided MSI workflows, we demonstrate this concept using arylsulfatase A-deficient (ARSA−/−) mice, a model of human metachromatic leukodystrophy (MLD). In these mice, sulfatides, a family of sulfated glycosphingolipids, selectively accumulate in kidneys and other organs[30–32]. Because of this well-understood biology of ARSA−/− mice, we could use renal

sulfatides, whose masses, chemical sum formulae and structures (but not their quantities) are known, as qualitative ground truth for statistical evaluation of QCL-MIR imaging-guided MSI methods, for method validation and for benchmarking against 4D LC-TIMS-MS sulfatide lipidomics, against previous sulfatide MSI studies and against CCS value prediction models[30,32–45].

In this work, we develop QCL-MIR imaging-guided MSI workflows and computational tools for spatially resolved deep lipidomics profiling and make them and extensive spatial lipidomics data available as a community resource. This concept enables QCL-MIR imaging-guided spatially resolved on-tissue $MS^2$ of lipids by iprm-PASEF using ion mobilograms that we optimize for best resolution for precursor ion selection. To demonstrate how the QCL-MIR imaging-guided MSI with iprm-PASEF workflow can support the investigation of new biology, we systematically characterize lipids that are upregulated in white matter lesions in spinal cords of mice suffering from experimental auto-immune encephalomyelitis (EAE), a clinically relevant model of human multiple sclerosis[46–48].

## Results

### Quantum cascade laser-based mid-infrared imaging microscopy to guide spatially focused data acquisition in MALDI imaging

Disentangling the molecular complexity of biological specimens by high performance liquid chromatography (HPLC) separation and thorough structural elucidation by $MS^2$ fragmentation analysis have been the hallmarks of LC-MS-based lipid/metabolite analysis for decades. However, such LC-MS-like analytical depth, sensitivity and confidence of identification is currently lacking in spatial MSI lipidomics.

As a solution to this conundrum, we proposed spending more MSI analysis time and depth on user-defined morphological structures of interest, i.e., on fewer pixels than those of entire tissue sections. Previously, we suggested FT-IR-based MIR vibrational tissue imaging for ROI definition, followed by MSI restricted to these ROIs[12]. However, in the past, such an FT-IR-based MIR imaging-guided MSI workflow lacked data acquisition speed and provided insufficient molecular specificity. It was therefore not very sensible to implement it in laboratories[49]. Meanwhile, QCL-based MIR instruments with a focal plane array detector offer scanning capabilities with microscopy quality and improved sensitivity due to high power density and much higher speed for acquisition of full spectra in the "fingerprint" region (950–1800 cm$^{-1}$; Fig. 1a(i)). The QCL-based MIR imaging microscope used within this study records 5 million $5 \times 5$ μm$^2$-sized pixels in 10 min (~8750 pixels per sec versus 50 pixels per sec with FT-IR imaging[12]) compared to 7 h for 175,000 $10 \times 10$ μm$^2$-sized pixels in TIMS-MSI. Available coherence reduction ensures effective suppression of sample-dependent phase shifts for structured tissues where the objects of interest, e.g., cells, have the same dimension as the wavelength of the light source (~5 μm)[50].

To maximize analytical depth per pixel, we set out to develop experimental workflows and IT tools for spatially focused MSI data acquisition on fewer pixels that are preselected by QCL-MIR imaging (Fig. 1a; Supplementary Fig. 1): As demonstrated for murine brain tissue sections, mid-infrared absorbance spectra were first recorded using indium tin oxide (ITO) glass slides (Supplementary Fig. 2). After data pre-processing, distinct spectral features were selected from the fingerprint region and corresponding ion images were used for image segmentation and definition of morphological regions-of-interest by, e.g., k-means clustering[25] (Fig. 1a(ii) and Fig. 1a(iii)). A separate single wavenumber (1656 cm$^{-1}$) whole-slide MIR reference image is co-registered with the hyperspectral dataset to generate the MSI data acquisition file, thus effectively enabling ROI-targeted MSI data acquisition with high analytical depth (Fig. 1a(iv) to Fig. 1a(vi)).

To validate this workflow, we first examined whether the photonic interaction of the QCL light with the tissue's molecules may cause lipid alterations[7]. However, no marked lipid changes (m/z 600–1700) were

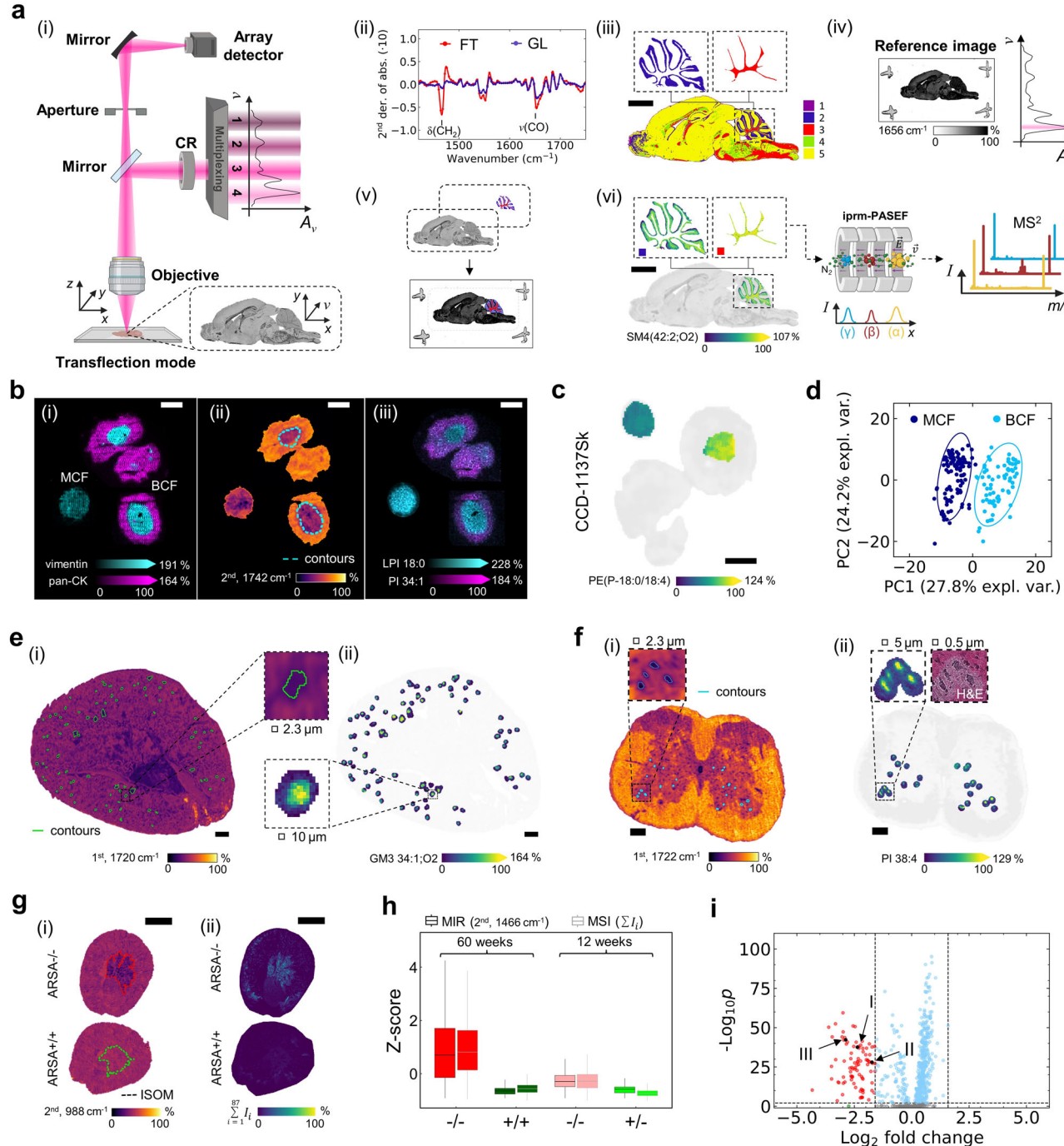

observed in murine brain tissue sections ($n = 4$ biological replicates) irradiated with the high-power laser for 15 min in either *single wavenumber* or *sweep scan* mode (Supplementary Figs. 3 and 4).

To ensure wide applicability of the QCL-MIR imaging-guided MSI approach in biomedical research, we further investigated five different biomedical examples ranging from macromorphological structures like kidney medulla or glomeruli down to the single cell level. First, we examined a two-cell types, i.e. two ROIs, in-vitro 3D-cell culture model of cancer-like aerobic glycolysis and reverse Warburg effect[51]. This model features biculture spheroids (300–500 µm diameter) consisting of two human cell lines, vimentin-positive CCD-1137Sk fibroblasts forming the core and pan-cytokeratin (pan-CK)-positive HT-29 colon cancer cells that engulf them, as confirmed by multiplex-MALDI-immunohistochemistry (IHC)[52] (Fig. 1b(i)). Analysis of hyperspectral MIR fingerprint data revealed the lipid-associated bands at

wavenumbers 1466 cm$^{-1}$ (CH$_2$ bending vibration) and 1742 cm$^{-1}$ (C=O vibration) as molecular features capable of distinguishing between the two cell types, as indicated by the precise match of the fibroblast core outlines determined by MALDI-IHC and MIR imaging (Fig. 1b(ii); Supplementary Fig. 5a). Consistent with recent observations in (brain) tumor patient samples where transcripts of glycerophospholipid (GPL) remodeling enzymes were overexpressed compared to surrounding non-tumor tissue[2], GPLs such as phosphatidylinositol PI 34:1 were more prominent in cancer cells, whereas lyso-GPLs, e.g. LPI 18:0, were more abundant in fibroblasts (Fig. 1b(iii)).

As a feasibility study, we compared monoculture fibroblast (MCF) spheroids with the QCL-MIR imaging-guided fibroblast core of bicultures (BCF), to investigate lipidomic reprogramming of BCF (compared to MCF) induced in the latter by being surrounded and separated from culture medium by cancer cells[53].

**Fig. 1 | QCL-MIR imaging to guide spatially focused data acquisition in MSI.**
**a** Overview of QCL-based MIR imaging microscope (i), and the QCL-MIR imaging-guided MSI workflow. $A_v$: absorbance, v: wavenumber, CR: coherence reduction. (ii) Selected spectral features ($2^{nd}$ derivative of absorbance ($2^{nd}$)) discriminating cerebellar fiber tracts (FT) and granular layer (GL): 1466 cm$^{-1}$ (CH$_2$ bending vibration) and 1742 cm$^{-1}$ (C = O vibration). (iii) Feature-selective image segmentation and ROI definition. (iv) Co-registration of *single wavenumber* (1656 cm$^{-1}$) reference image (v) with hyperspectral dataset (i). (vi) Cerebellar ROI-focused MSI using iprm-PASEF. Scale bar, 2 mm. BioRender. https://BioRender.com/ymashbv. **b** Multimodal comparison of CCD−1137Sk fibroblast/HT-29 cancer biculture (BCF) versus monoculture fibroblast (MCF) spheroids. (i) Ion images of multiplex-MALDI-IHC[52] using anti-vimentin (*m/z* 1230.84; fibroblast) and anti-pan-CK (*m/z* 1288.71; cancer) antibodies. Mass window ±10 ppm. (ii) MIR data (1742 cm$^{-1}$; $2^{nd}$) of spheroid section, fibroblast core outline (cyan, dashed) derived by clustering (k = 2) of MALDI-IHC data. (iii) Ion images for *m/z* 835.54 (PI 34:1[M-H]$^-$; fibroblasts) and *m/z* 599.32 (lyso-PI 18:0[M-H]$^-$; cancer). Scale bar, 200 μm. **c** QCL-MIR imaging-guided ion image of *m/z* 722.51 (PE(P-36:4[M-H]$^-$). Scale bar, 200 μm. **d** Principal component analysis (PCA) of MSI data distinguishes MCF (dark blue) and BCF (cyan). **e** (i) MIR image of

ARSA−/− mouse kidney (1720 cm$^{-1}$; $1^{st}$) with putative glomerular ROI (green outline). (ii) QCL-MIR imaging-guided ion image of *m/z* 1151.71 (GM3 34:1;O2[M-H]$^-$; mass window ±10 ppm) with highlighted glomerular ROI. Scale bar, 300 μm. **f** (i) MIR image of EAE mouse spinal cord (1722 cm$^{-1}$; $1^{st}$) with putative motor neuron ROI (blue outline). (ii) QCL-MIR imaging-guided ion image of *m/z* 885.549 (PI 38:4[M-H]$^-$; mass window ±10 ppm) with highlighted neuron ROI. Scale bar, 200 μm. **g** Sulfatide accumulation in ARSA−/− mice by (i) MSI (sum intensity distribution of 87 sulfatides[32]), and (ii) MIR imaging at 988 cm$^{-1}$ ($2^{nd}$, C$_\beta$-O vibration). Superimposed kidney inner stripe of outer medulla (ISOM) ROI determined by clustering of MSI data (red and green dashed lines). Scale bar, 2 mm. **h** Box-plots of Z-score values reveal lipid accumulation in the ISOM region by both MSI and MIR imaging (1466 cm$^{-1}$; $2^{nd}$) for n = 4 biological replicates. Boxplots indicate median (middle line), 25$^{th}$ and 75$^{th}$ percentile (box) and whiskers (1.5 times the interquartile range). i Volcano scatter plot of qTOF-MSI data for ARSA−/− vs. ARSA+/+ reveals accumulation of sulfatides[32] for ARSA−/− (red dots), e.g., I) SM4 34:1;O2[M-H]$^-$, II) SM4 38:1;O3[M-H]$^-$, and III) SM3 42:1;O2[M-H]$^-$. Statistical significance was tested by two-sided standard t-test. P-values are Benjamini−Hochberg-corrected. Source data is provided as Source Data file.

---

Principal component analysis on 105 MCF- and 72 BCF spheroids (technical replicates from n = 7 wells for BCF and n = 11 wells for MCF, individually labelled in Supplementary Fig. 6e) of MSI data indicated unique lipidomic profiles for both 3D-cellular systems (Fig. 1d). Chemometrics- and machine learning-based feature extraction (LASSO regression) revealed lipid candidates, e.g., phosphatidylethanolamine PE(P-36:4), with different abundance in BCF than MCF, which were identified using iprm-PASEF analysis. (Fig. 1d; Supplementary Figs. 6, 7; Supplementary Tables 1-9).

Next, we explored small tissue morphologies like glomeruli in murine kidneys to validate the MIR-based hyperspectral tissue segmentation further. Gangliosides as markers of these functional filtration units are well-characterized, and autofluorescence-directed MSI has recently been used for their detailed molecular analysis[42]. Using ganglioside GM3 34:1;O2 (both negative mode ion images and molecular probabilistic maps (MPMs)[2]) as a marker, we compared MALDI-qTOF-MSI and MIR imaging for entire dried kidney cryosections (n = 4 biological replicates) by correlative MSI-MIR imaging (Supplementary Fig. 8ab). Utilizing the spectral region at around 1720 cm$^{-1}$ ($1^{st}$ derivative of transmittance) between the amide I and carbonyl vibrations allowed for discrimination between glomerular structures and surrounding kidney cortex. Additionally, ROIs were chosen such that the number of objects identified by both MIR imaging and MSI compared to the latter alone was maximized, thus rather accepting false-negatives, but avoiding false-positive detections in MIR imaging (Fig. 1e; Supplementary Fig. 8c). On average, more than 85% of MIR-defined ROIs contained glomeruli, as defined by GM3 34:1;O2 presence (Supplementary Fig. 9a−d). To compensate for possible errors in image co-registration, MIR-defined ROIs were computationally expanded. To evaluate the magnitude of these effects, the Euclidean distance between the centers of the expanded MIR-defined ROIs and the GM3 34:1;O2-defined MSI areas determined the off-set. Averaged across all glomeruli in four independent data sets, the spatial shifts were 21 ± 4 μm compared to MIR (i.e., the taught reference image) and MSI pixel sizes of 2.3 μm and 10 μm, respectively (Supplementary Fig. 9ef). Focusing MSI data acquisition on these glomeruli-containing ROIs instead of full tissue MSI reduced data acquisition time by > 95% (8483 pixels instead of 216,411). This allowed for subsequent redirection of time and effort into very-high-confidence identification of ten gangliosides in these ROIs by ion mobility-based on-tissue fragmentation using iprm-PASEF in TIMS-MSI, validated by the additional analysis of a ganglioside standard (Supplementary Fig. 9g−k; Supplementary Table 10).

As the final test of the workflow's versatility, we examined whether ROIs as small as single cells could be defined. To this end, individual neuron-containing ROIs that could, for instance, be further examined in single cell studies in motor neuron disease were identified by QCL-MIR imaging microscopy in the gray matter of frozen murine spinal

cord sections (n = 2 biological replicates) and then computationally expanded (Fig. 1f(ii); Supplementary Figs. 10a and 11). Distinct motor neuron locations were characterized by the spatial distribution of the phosphatidyl inositol PI 38:4 and validated by a hematoxylin and eosin (H&E) image of the same section (Fig. 1f(ii)).

## Genetically engineered mice with defined metabolic alterations as a qualitative ground truth in MSI method development and validation

Using our QCL-MIR imaging-guided MSI toolbox, we endeavored to chemically characterize an entire lipid class, sulfatides, as comprehensively and completely as possible, since it is neither sufficiently covered in public databases (SwissLipids knowledgebase, https://swisslipids.org/; LIPID MAPS Structure Database, https://www.lipidmaps.org/) nor in instrument vendor software. Furthermore, we aimed to provide methods for iprm-PASEF with tailored ion mobility separation, i.e. using long ramp times, to analyze MIR-defined ROIs and to generate an extensive comparative MSI- and LC-TIMS-MS data resource for the MSI and lipidomics communities.

To this end, we introduce the concept of using knock-out mice to approach analytical ground truths in complex tissue analytics. Therefore, we used kidneys of arylsulfatase A-deficient mice (ARSA−/−) that we first analyzed in 2011 using low-resolution MALDI-TOF MSI incapable of accurately identifying lipids[32] (Supplementary Table 11). In kidneys of ARSA−/− mice, sulfatides of the SM4, SM3, SM2a, SM1a/b, and SB1a subclasses accumulate primarily in the medulla and papilla[30–32,54] (Fig. 1g−i; Supplementary Fig. 12 for sulfatide metabolism), where they are known to be critical for urinary pH and ammonium excretion and have been characterized at single intercalated cell level[39,43]. In the brain, they promote neurodegeneration in MLD. Sulfatide accumulation was observed consistently across MSI and MIR imaging (Supplementary Fig. 13). The inner stripe of the outer medulla (ISOM) and the inner medulla/papilla (IMP) were readily obtained by image segmentation utilizing a well-defined subset of spectral features in the "fingerprint" region, i.e., the lipid-associated CH$_2$ bending vibration at 1466 cm$^{-1}$ and the C$_\beta$-O vibration of the 3-sulfogalactosyl head group at 988 cm$^{-1}$ (Supplementary Figs. 5b and 14). Comparison of segmentations based on each imaging modality and with reference histology suggested that MIR imaging did not alter coverage of the ISOM and IMP ROIs (Supplementary Fig. 15).

## QCL-MIR imaging-guided trapped TIMS-MSI enables extensive fragmentation analysis on tissue and sulfatide identifications on par with LC-based 4D-TIMS-PASEF

TIMS-TOF-MSI, in principle, offers substantial capabilities for deep lipidomics profiling[55,56]. However, as the use of TIMS separation

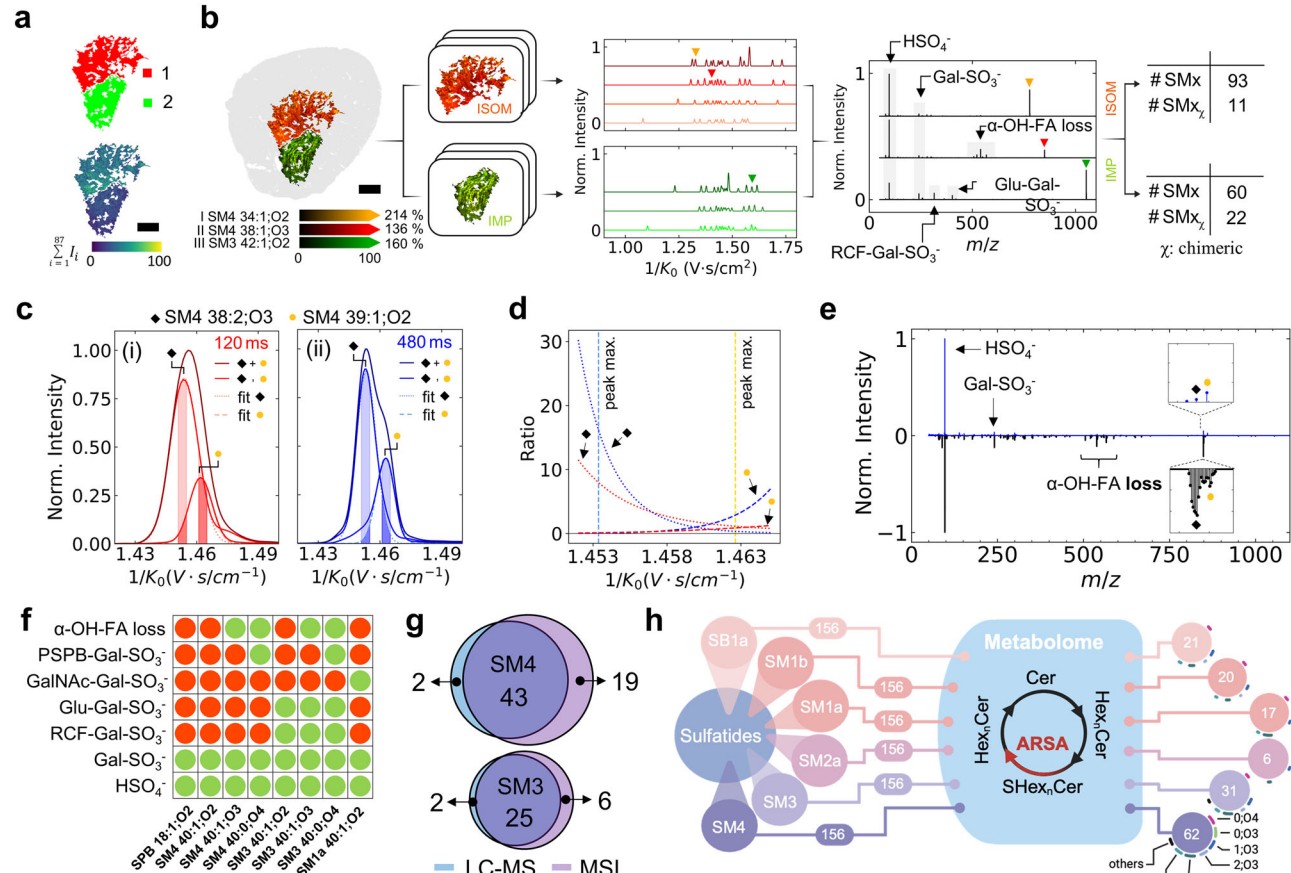

**Fig. 2 | Deep sulfatide profiling via QCL-MIR imaging-guided TIMS-MSI in ARSA −/− mouse kidney. a** ISOM (red, 1) and inner medulla/papilla (IMP; green,2) identified as ROIs via QCL-MIR imaging (top) and sulfatide distribution in MSI (sum intensities; bottom). Scale bar, 500 μm. Partly created in BioRender. https://BioRender.com/spcji05. **b** Overlaid QCL-MIR imaging-guided ion image of **I)** *m/z* 778.515 (SM4 34:1;O2[M-H]⁻, orange), **II)** *m/z* 850.572 (SM4 38:1;O3[M-H]⁻, red), and **III)** *m/z* 1052.692 (SM3 42:1;O2[M-H] ⁻; green) within kidney (grey). Mass window ±3 ppm. Subsequently, precursor ions for iprm-PASEF are region-selectively filtered for on-tissue fragmentation analysis to reveal molecular identities. In total, we acquired iprm-PASEF-derived MS² spectra for 153 sulfatide species of which 33 are considered chimeric. Scale bar, 500 μm. **c** Separation of even-and odd-chain sulfatides on-tissue: Extracted ion mobilograms and iprm-PASEF isolation windows for SM4 38:2;O3 (black rhombus) and SM4 39:1;O2 (orange dot) acquired with (**i**) 120 ms (red) and (**ii**) 480 ms TIMS ramp time (blue). Gaussians describe the data for each molecule (dashed and dotted lines). **d** Peak intensity ratios within the isolation window (filled areas under curve in (**c**)) for both sulfatides and ramp time settings. Peak intensity maxima are marked in dotted lines. **e** Butterfly plot of iprm-PASEF-derived MS² spectra (top, blue) and conventional on-tissue MS² without ion mobility separation (bottom, black) for *m/z* 848.591 (SM4 39:1;O2[M-H]⁻). Fragments at *m/z* 568.28, *m/z* 540.29, and *m/z* 522.28 refer to the loss of the α-OH-FA. **f** MS² fragments observed (green dots) in iprm-PASEF analysis of selected sulfatides (columns). **g** Venn diagram of sulfatide subclasses identified by 4D-LC-TIMS-MS (blue) and MALDI-TIMS-MSI (purple). **h** Evaluation of the ground truth concept: Total number of identified sulfatides per subclass out of 156 theoretical sulfatide configurations. Created in BioRender. https://BioRender.com/64y1bin. Source data is provided as a Source Data file.

increases MSI data acquisition time up to 10-fold, in practice, most imaging studies refrain from using the TIMS capabilities and are conducted in qTOF/TIMS-off mode instead.

Hence, we investigated the benefit in deep spatial sulfatide profiling achieved by in-depth analysis of smaller ARSA−/− kidney ROIs (ISOM and IMP) defined by QCL-MIR imaging (compare Fig. 2a). We comprehensively compared conventional qTOF-mode MSI of whole kidney slices (including on-tissue MS² based on *m/z* values for precursor selection alone) with QCL-MIR imaging-guided analysis focused to the ISOM and IMP ROIs but using TIMS-MSI with iprm-PASEF (Fig. 2b, Supplementary Fig. 16). For example, in non-guided whole kidney qTOF-mode MSI, 54,100 pixels were measured in 1:22 h; in QCL-MIR imaging-guided TIMS-MSI 9,600 pixels were assessed in 1:19 h (both 20 μm lateral step size). For iprm-PASEF at 40 μm lateral step size 883 pixels were investigated in IMP (8 min) and 1554 pixels in ISOM (13 min). To achieve and validate highest levels of confidence in fragment annotation and subsequent sulfatide identification, we first confirmed that on-tissue iprm-PASEF MS² data were in strong agreement with results obtained with brain sulfatides standard mixture

(Supplementary Fig. 17). Furthermore, MS² spectra obtained for kidneys from *n* = 4 different mice demonstrated highest degrees of similarity ( > 0.99), as judged by cosine similarity of three example sulfatides (Supplementary Fig. 18). Generally speaking, TIMS-MSI with iprm-PASEF requires baseline separation of sulfatides in the extracted ion mobilogram (EIM). Effective separation of almost isobaric even-chain and odd-chain sulfatides, *m/z* 848.557 (SM4 38:2;O3[M-H]⁻) and *m/z* 848.591 (SM4 39:1;O2[M-H]⁻), required an extended ramp time of 480 ms (Fig. 2c). Feasibility of subsequent iprm-PASEF analysis was evaluated by comparison of area-under-curve (AUC) ratios for the respective EIMs within the targeted isolation window (Fig. 2d). A ramp time of 480 ms led to a two-fold increase in the AUC ratio for SM4 38:2;O3[M-H]⁻, and a three-fold increase for SM4 39:1;O2[M-H]⁻. The iprm-PASEF-derived MS² spectrum of *m/z* 848.591 (SM4 39:1;O2[M-H]⁻) enabled the unequivocal identification of this odd-chain sulfatide (OCS). In contrast, without ion-mobility-based precursor isolation (i.e., by conventional on-tissue MS²), this MS² spectrum is obscured by more intense fragments of *m/z* 848.557 (SM4 38:2;O3[M-H]⁻), highlighted by characteristic fragments resulting from α-hydroxy fatty acid loss, and

**Table 1 | Cumulative numbers of sulfatide subclass isoforms identified in ARSA−/− mouse kidney by LC-MS or MALDI-MSI using various analytical workflows**

| | | | | SM4 | SM3 | SM2a | SM1b | SM1a | SB1a | total |
|---|---|---|---|---|---|---|---|---|---|---|
| LC-TIMS-MS | timsTOF-MS | | 60w_1 | 41 | 25 | 0 | 0 | 0 | 26 | 92 |
| | | | 60w_2 | 42 | 26 | 0 | 0 | 0 | 27 | 95 |
| | | | 12w_1 | 25 | 14 | 0 | 0 | 0 | 4 | 43 |
| | | | 12w_2 | 33 | 20 | 0 | 0 | 0 | 11 | 64 |
| | timsTOF-MS[2] | | 60w_1 | 38 | 25 | 0 | 0 | 0 | 26 | 89 |
| | | | 60w_2 | 39 | 26 | 0 | 0 | 0 | 27 | 92 |
| | | | 12w_1 | 22 | 14 | 0 | 0 | 0 | 4 | 40 |
| | | | 12w_2 | 30 | 20 | 0 | 0 | 0 | 11 | 61 |
| MALDI timsTOF MSI | qTOF mode | all | 60w_1 | 44 | 30 | 4 | | 13 | 10 | 101 |
| | | | 60w_2 | 44 | 33 | 5 | | 17 | 13 | 112 |
| | | | 12w_1 | 34 | 21 | 2 | | 11 | 9 | 77 |
| | | | 12w_2 | 30 | 25 | 1 | | 9 | 9 | 74 |
| | | "clean" | 60w_1 | 15 | 16 | 2 | | 10 | 4 | 47 |
| | | | 60w_2 | 14 | 22 | 2 | | 12 | 5 | 55 |
| | | | 12w_1 | 11 | 13 | 1 | | 8 | 4 | 37 |
| | | | 12w_2 | 10 | 16 | 1 | | 6 | 3 | 36 |
| | MIR-guided | all | 60w_1 | 60 | 31 | 6 | 20 | 17 | 21 | 155 |
| | | | 60w_2 | 62 | 31 | 6 | 20 | 17 | 20 | 156 |
| | | | 12w_1 | 52 | 26 | 6 | 13 | 11 | 12 | 120 |
| | | | 12w_2 | 51 | 26 | 6 | 14 | 12 | 17 | 126 |
| | | "clean" | 60w_1 | 40 | 19 | 6 | 17 | 17 | 16 | 115 |
| | | | 60w_2 | 44 | 23 | 6 | 15 | 17 | 14 | 119 |
| | | | 12w_1 | 34 | 15 | 6 | 13 | 11 | 11 | 90 |
| | | | 12w_2 | 36 | 12 | 6 | 13 | 12 | 12 | 91 |

Whole kidney sections of 12- or 60-week-old ARSA−/− mice ($n = 2$ each) were analyzed by LC-TIMS-MS or –MS[2] and MALDI-MSI (TIMS/qTOF). "Clean" peaks fulfilled higher quality standards, defined as peaks that do not feature at least one other peak within a mass window of ±1.1 Da (qTOF mode MSI) and a mobility window of ±0.005 Vs/cm² (TIMS-MSI), and were manually curated. QCL-MIR imaging-guided MSI focused on the kidney's ISOM and IMP ROIs only.

concomitant inability to identify SM4 39:1;O2[M-H]⁻ (Fig. 2e; Supplementary Fig. 19c, d). Extensive iprm-PASEF analysis led to a detailed atlas of characteristic fragments for the sulfatide subclasses (Fig. 2f; Supplementary Figs. 20–23). MALDI-MSI was at least on par with 4D-LC-TIMS-MS lipidomics in this comparison. Relying on similar ion mobility settings, the numbers of SM4 and SM3 sulfatides identified with the two methods were on par (Fig. 2g). Aiming for maximal coverage, in addition we identified complex SM2 and SM1 sulfatides by TIMS-MSI using iprm-PASEF with adjusted TIMS settings (Supplementary Fig. 24). In two 60-week-old mouse kidneys we elucidated 101 and 112 sulfatides by TIMS-MSI in qTOF mode and 155 and 156 by QCL-MIR imaging-guided TIMS-MSI (Table 1). Six sulfatide subclasses were detected in method-specific manner (Fig. 2h). SB1a isoforms were detected as [M+Na-2H]⁻ adducts in TIMS-MSI[57]. Sulfatides such as SM1a/b predominantly form doubly charged ions in 4D-LC-TIMS-MS, and their detection within the set ion mobility and $m/z$ range is feasible, given also that no in-source decay was evidenced for any sulfatides. However, SM1a/b were not detected in 4D-LC-TIMS-MS. The origin of these species in TIMS-MSI-whether due to in-source-decay of SB1a, degradation during sample preparation for MALDI, or endogenous occurrence remains unclear. Nevertheless, their value for structure-CCS-relationship analysis is obvious. Structure elucidation based on iprm-PASEF spectra allowed for precise localization of the sulfate group and unequivocal classification of the SM1a/b isoforms, thus highlighting a major advantage of TIMS-MSI with iprm-PASEF. For SB1a, we identified the sodiated sulfate group to be predominantly located at the terminal galactose. In total, we benchmarked 157 sulfatides in the ARSA−/− mouse model out of 936 theoretical structural configurations as the ground truth (Fig. 2g, h; Table 1; Supplementary Tables 12 and 13; Supplementary Data 1–3). Our findings validate the

use of mutant mice with known patterns of metabolite accumulation in distinct tissue regions as ground truth in spatial lipidomics/metabolomics.

To aid statistical quality assessment, we defined "non-clean" sulfatide peaks for $m/z$ spectra in qTOF-mode MSI as the ones that featured at least one other peak within a mass window of ±1.1 Da that exceeded the intensity of the first sulfatide isotope peak (Supplementary Fig. 19a); the reason being that this interfering peak would prevent unequivocal precursor ion selection and on-tissue MS[2] sulfatide identification in "conventional" qTOF mode. In TIMS-MSI operation, peaks were classified as "non-clean" if the spectra contained a peak within a mass window of ±1.1 Da and a mobility window of ±0.005 Vs/cm². All peaks were manually classified as "clean" or "non-clean" (Table 1; Supplementary Fig. 19a; Supplementary Data 1–3). Of the 101 and 112 sulfatides seen in qTOF-mode MSI and the 155 and 156 detected in TIMS-MSI, about half (47 and 55) and about 75% (115 and 119) were considered as "clean", respectively (Table 1).

QCL-MIR imaging-guided TIMS-MSI with iprm-PASEF enabled new insights in deep spatial lipidomics, e.g., the spatial profiling of odd-chain sulfatides (OCS), a subgroup that is still underexplored. Odd-chain fatty acids (FAs) can be formed by elongation of gut-/microbiome-derived propionyl-CoA but also via α-oxidation by 2-hydroxy acyl-CoA lyase in mammalian cells[58,59]. The example of $m/z$ 848.557 (SM4 38:2;O3[M-H]⁻) and $m/z$ 848.591 (SM4 39:1;O2[M-H]⁻) demonstrated that non-resolved ("non-clean") odd-/even-chain sulfatides could be separated by ion mobility spectrometry and subsequently identified by iprm-PASEF analysis (Fig. 2c–e; Supplementary Figs. 19 and 20). The ability to unequivocally identify odd-chain membrane lipids also applied to PI isomers like PI 33:1 in the spheroid example (Supplementary Fig. 25).

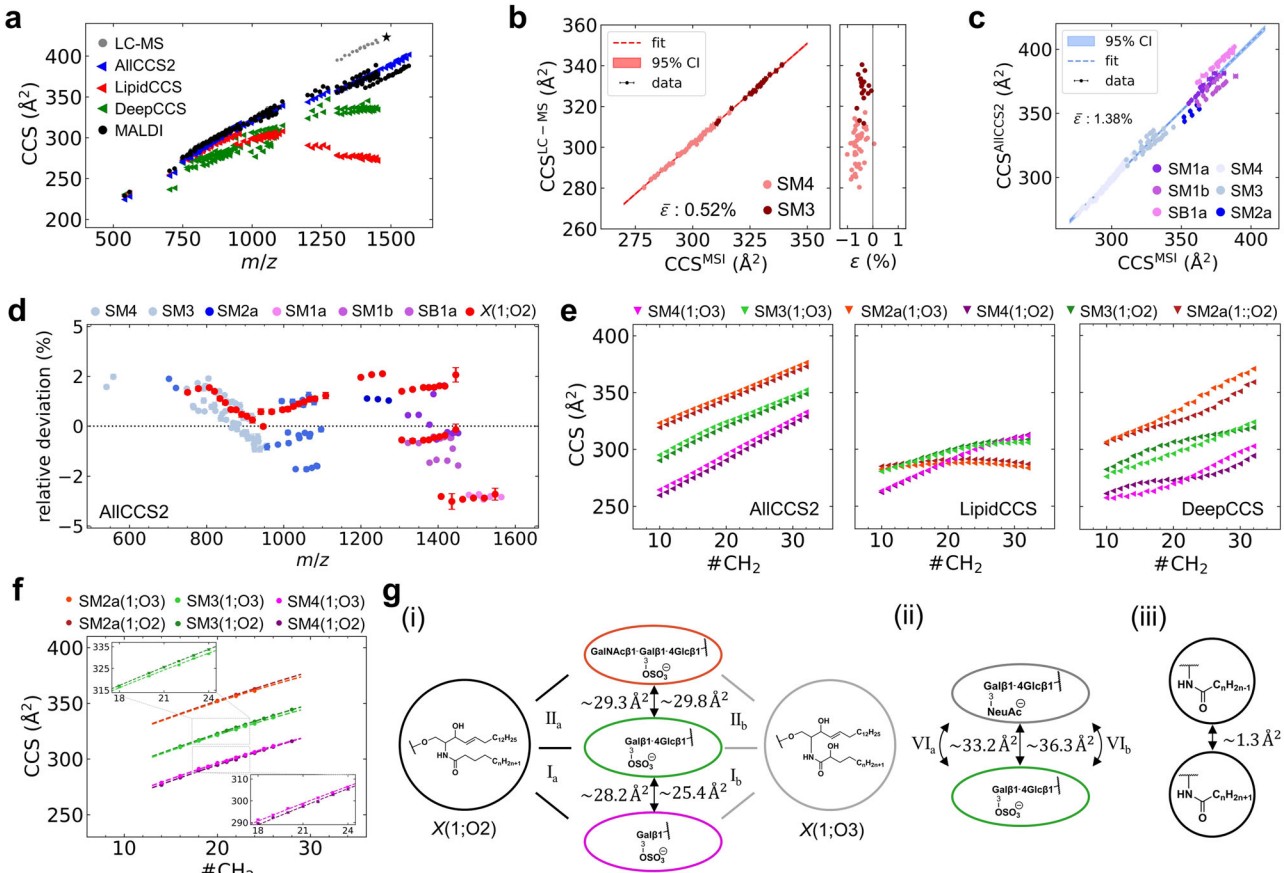

**Fig. 3 | Structure-CCS-relationships for sulfatides. a** Comparison of experimental (gray dots: LC-TIMS-MS; black dots: MALDI-TIMS-MSI) and predicted CCS values, modeled by LipidCCS[61] (red triangle), AllCCS2[34] (blue triangle) and DeepCCS[36] (green triangle). LC-MS-derived CCS values of SB1a[M-2H]2- /SB1a[M-HSO3]- are marked with a star. **b** Strong correlation (R2 = 0.9988; linear fit (red); 95% confidence interval (CI)) of LC-MS-derived and MALDI-MSI-derived ion mobility data. Mean relative deviation $\bar{\varepsilon}$ = 0.5%. **c** Correlation of experimental (MALDI-MSI) and predicted (AllCCS2) CCS values. Mean relative deviation $\bar{\varepsilon}$ is 1.4%. **d** Relative deviation $\varepsilon$ reveals inconsistent deviations per subclass of the predicted CCS values using AllCCS2 against CCS values obtained by MALDI-MSI. Uncertainties were derived from for $n$ = 4 biological replicates and expressed as standard deviation (Supplementary Table 13). **e** CCS values of

sulfatide subclasses as a function of fatty acid (FA) chain length for three different prediction tools, AllCCS2, LipidCCS and DeepCCS yielding ambiguous relative structural relationships. **f** Experimental CCS values and 2nd order polynomial fit to evaluate the contribution of the degree of glycosylation in the sulfated head group and the α-hydroxylation of the N-acyl FA. The relative positions are highlighted in the insets. **g** Visualization of structural relationships for the data presented in **f** (**i**), relative to ganglioside series (**ii**) and the contribution of saturated FA and mono-unsaturated FA. Surprisingly, the difference between SM3(O3) and SM4(O3) is reduced compared to the non-α-hydroxylated counterpart suggesting that interaction between the α-OH group and the head group may influence the three-dimensional structure. Source data is provided as a Source Data file.

Generally, QCL-MIR imaging-guidance to ISOM and IMP ROIs allowed for extensive iprm-PASEF analysis and structure elucidation of odd- and even chain-sulfatides directly on-tissue (Fig. 2d; Supplementary Fig. 20–23). Surprisingly and in contrast to even-chain sulfatides like SM4 38:2;O3[M-H]-, ion intensities for OCS like SM4 39:1;O2[M-H]- were unaltered in ARSA−/− kidneys, as indicated by ion images and a volcano plot (Supplementary Fig. 20a). As iprm-PASEF analysis unequivocally identified OCS, this finding may suggest that odd-chain galactosylceramides and lactosylceramides can be sulfated by cerebroside sulfotransferases (Supplementary Fig. 12), but that an arylsulfatase other than ARSA may catalyze their de-sulfation. This observation may be pursued in a separate study, as arylsulfatases constitute a growing enzyme family whose functions are not fully characterized yet[60]. In similar fashion, large numbers of sulfatides could be structurally elucidated by TIMS-MSI with iprm-PASEF (Fig. 2g; Supplementary Fig. 22 and 23, Supplementary Table 13).

## Structure-CCS-relationships based on experimental deep profiling spatial lipidomics data

This vast resource of spatial MALDI-MSI and corresponding LC-MS data, both acquired on TIMS-MS platforms, permitted deep inquiries

into the (spatial) sulfoglyco- lipidome. First, experimental CCS values (LC-MS and MSI) differed profoundly from those predicted by recent models LipidCCS, DeepCCS and AllCCS2[34,36,38,61] (Fig. 3a). In contrast, experimental CCS values were very consistent and independent of ion source and instrument usage at different sites (Mannheim and Mainz; R2 = 0.9988; mean relative deviation $\bar{\varepsilon}$ of 0.5%; Fig. 3b; Table 2; Supplementary Tables 12 and 13; Supplementary Data 4). LipidCCS was accurate ($\bar{\varepsilon}$ of 1%) for SM4, but inaccurate for all other sulfatide subclasses, whereas DeepCCS predicted all subclasses with a mean relative deviation of 8%. AllCCS2 yielded the most accurate prediction ($\bar{\varepsilon}$ of 1.4% for all subclasses, but, e.g., 2.48% for SM2a; Fig. 3ac; Supplementary Fig. 26).

Relative deviations were neither a function of subclass nor of FA chain lengths (Fig. 3d), as highlighted by the inconsistent behavior of the relative error of the subclass X(1:O2) predicted by AllCCS2. These discrepancies between prediction and experiment, revealed by QCL-MIR imaging-guided TIMS-MSI together with an ambiguous trend in the relative position of the homologous series of subclasses across various prediction tools (Fig. 3e; Supplementary Fig. 27), prompted us to analyze structure-CCS-relationships (SCR) in more detail.

**Table 2 | Relative values for structure-CCS-relationships of sulfatide subclasses**

| | a: X(1;O2) | b: X(1;O3) |
|---|---|---|
| I: ΔCCS(SM4, SM3) | 28.2 ± 0.1 | 25.4 ± 0.2 |
| II: ΔCCS(SM3, SM2a) | 29.3 ± 0.1 | 29.8 ± 0.4 |
| III: ΔCCS(SM2a, SM1b) | 12.8 ± 0.4 | 15.3 ± 0.9 |
| IV: ΔCCS(SM1b, SM1a) | 7.5 ± 0.5 | 4.2 ± 1.5 |
| V: ΔCCS(SM1a, SB1a) | −1.2 ± 0.5 | 2.0 ± 1.3 |
| VI: ΔCCS(SM3, GM3) | 33.2 ± 0.2 | 36.3 ± 0.2 |
| | c: X(0;O3)[1] | d: X(0;O4) |
| VII: ΔCCS(SM4, SM3) | 22.2 ± 0.2 | 20.9 ± 0.4 |
| | e: X(1;O2)–X(2;O2) | f: X(1;O3)–X(2;O3) |
| VIII: ΔCCS(SM4) | 1.3 ± 0.2 | 1.3 ± 0.2 |
| IX: ΔCCS(SM3) | 1.3 ± 0.3 | 1.3 ± 0.2 |
| | g: X(1;O2)–X(0;O3) | h: X(0;O3)–X(0;O4) |
| X: ΔCCS(SM4) | 3.9 ± 0.2 | 2.0 ± 0.1 |
| XI: ΔCCS(SM3) | 2.9 ± 0.2 | 1.7 ± 0.3 |

Relative differences in CCS (mean for $n = 4$ biological replicates) values were obtained by a parallel line model using a $2^{nd}$ order polynomial fit. 1: The relative difference was obtained based on TIMS-MSI and LC-TIMS-MS data. Therefore, the latter was shifted by the mean relative deviation. All CCS values are presented in Å².

To compare series of sulfatides in structural subclasses such as the α-hydroxylated (X 18+n:1;O3) and non-α-hydroxylated (X 18+n:1;O2) sulfatides, experimental CCS of a homologous series were better modeled by $2^{nd}$ order polynomial fits than by linear fits or $y = b_{CCS}x^{2/3} + a_{CCS}$ fits that have been used to describe the progression of CCS values of polymers[62] (Supplementary Fig. 28). First, we assessed the contribution of glycosyl head groups and chain lengths of N-acyl-linked FA to experimental CCS values for both the α-hydroxylated (X 18+n:1;O3) and non-α-hydroxylated (X 18+n:1;O2) sulfatides compared to predicted CCS (Fig. 3f; Table 2).

Only AllCCS2 correctly predicted the strictly monotonous increase with FA chain length, but it did not predict a subtle difference between the O2- and O3-subseries: In both MSI and LC-MS data (Supplementary Fig. 29), $CCS_{O2} > CCS_{O3}$ was observed for complex SM2a and SM3 sulfatides, but not for SM4, whereas inconsistent trends were observed for different prediction tools. Especially the most accurate predictor investigated here, AllCCS2, showed $CCS_{O3} > CCS_{O2}$ for all classes. Employing a parallel line model (see Methods), we determined constant differences ΔCCS between SM2a and SM3 isoforms of identical FA chain length of 29.3 ± 0.1 Å² and 29.8 ± 0.4 Å² for the O2- and O3-subseries, respectively (Fig. 3f and 3g; Supplementary Fig. 30). However, ΔCCS between SM3 and SM4 isoforms was 28.2 ± 0.1 Å² and 25.4 ± 0.2 Å² for the O2- and O3-subseries, respectively (Fig. 3g(i); Supplementary Fig. 30).

This significant difference suggests that an interaction between the α-OH-group and the glycosyl head group may influence the three-dimensional structure. The finding is also supported by our LC-TIMS-MS data (Supplementary Fig. 29; Supplementary Table 12). This effect seems to be reversed for lipid classes with unrelated glycosyl head groups like sulfatides and GM3 gangliosides (Fig. 3g(ii), Table 2, Supplementary Fig. 31). In contrast, single sites of FA unsaturation that, because of the cis-configuration of double bonds in FA, introduce a kink in their three-dimensional structure leading to a reduction of the CCS values by 1.3 ± 0.2 Å² (Fig. 3g(iii), Supplementary Fig. 32). The position of the double bond was experimentally not determined. Our data indicated that phytosphingoid base-containing sulfatides exhibit the same trend as the SM3 and SM4 isoforms for the O2- and O3-subseries, i.e., that the relative difference in CCS is reduced for X(0;O4) compared to X(0;O3) (Supplementary Fig. 33). Overall, we provide a comprehensive overview of the CCS behavior across six different sulfatide classes (Table 2; Supplementary Fig. 30).

## Lipid remodeling in experimental autoimmune encephalomyelitis mice

To demonstrate how QCL-MIR imaging-guided TIMS-MSI with iprm-PASEF can drive biomedical discoveries, we investigated the dynamic lipid remodeling in white matter of mouse spinal cord at peak disease instigated by experimental autoimmune encephalomyelitis (EAE), a model of human multiple sclerosis[46–48]. MIR images highlighted lesions in the white matter of EAE but not healthy spinal cords, suggesting extensive remodeling with substantial changes in lipid composition in the former (Fig. 4 (i)). These lesions were readily identified and defined as ROIs (Fig. 4b (ii); Supplementary Fig. 10). To define white matter of control spinal cords as reference ROI, we computationally shrunk the white matter segment defined by QCL-MIR imaging to match the size of the combined EAE lesion ROIs (Fig. 4a (ii); Supplementary Figs. 10 and 34). By focusing exclusively on the QCL-MIR-imaging-derived lesion ROIs, we reduced the measurement time by up to 20-fold (depending on the lesion size), compared to the measurement time for an entire spinal cord tissue section at 5 μm pixel size.

Volcano scatter plot analysis across $n = 3$ biological replicates revealed upregulated m/z features, notably I m/z 616.472 (ceramide-1-phosphate CerP 34:1;O2[M-H]⁻), a signaling lipid, II m/z 687.545 (CerPE 36:1;O2[M-H]⁻), as well as the cell membrane lipid m/z 885.549 (PI 38:4[M-H]⁻), in EAE lesions (Fig. 4c), validated by subsequent iprm-PASEF MS² analysis (Supplementary Fig. 35). It is tempting to speculate that moderate upregulation of PI 38:4 may be proinflammatory, as phospholipase A2 cleavage of this could trigger release of the eicosanoid precursor arachidonic acid. Compounds I and II are strikingly more than 25-fold enriched in sclerotic lesions, and multiple additional compounds displayed enrichments of 4- to 20-fold in lesions (Fig. 4c). Ceramide-1-phosphate (compound I) has already been associated with myelin sheath degenerative pathogenesis. Interestingly, deletion of ceramide kinase and concomitant reduction in CerP levels ameliorates disease progression in the cuprizone mouse model of myelin degeneration and structural rebuilding[63]. LC-MS lipidomics of human post mortem multiple sclerosis white matter tissue suggested that CerP could be a candidate biomarker of the progressive phase[64], but its localized enrichment in active lesions of EAE has not been demonstrated yet. In contrast, very little is known about the function of ceramide phosphoethanolamines (CerPE) that are synthesized by sphingomyelin synthase-related (SMSr) enzymes[65]. In Drosophila, CerPE maintains synaptic glutamate homeostasis, the dysregulation of which (i.e., glutamate excitotoxicity) is a feature of multiple sclerosis and EAE[66]. Hence, 25-fold enrichment of CerPE and its localization in EAE lesions suggests that this class of lipids should be studied in more detail (Fig. 4d). Combining H&E staining with MIR and MSI images provides a multi-faceted view of the histological and lipidomic changes in Ctrl and EAE spinal cords at peak disease. The ion images for I, and II and PI 38:4 highlight their significantly increased ion intensity and thus presumably accumulation within EAE lesions (Fig. 4d and Supplementary Fig. 34). Detailed t-statistical analysis for I and II and PI 38:4 across $n = 3$ biological replicates highlight the importance of these features (Fig. 4e; Supplementary Figs. 36 and 37). Structural confirmation was achieved by on-tissue iprm-PASEF analysis (Fig. 4f; Supplementary Fig. 35). Taken together, visualization of CerP 34:1;O2 and CerPE 36:1;O2 in EAE lesions and strong increases in their ion intensity compared to control white matter together with on-tissue MS² structure elucidation demonstrate how the QCL-MIR imaging-guided MSI workflow can support the discovery of new biology. These findings not only enhance our understanding of localized molecular alterations, here: sphingolipids[64], in demyelinating diseases, but also point to potential biomarkers and therapeutic targets for multiple sclerosis and related conditions[64]. The example of fingolimod (FTY720), an S1PR antagonist that blocks the inflammatory actions of the related signaling lipid sphingosine-1-phosphate (S1P) suggests that analogous CerP signaling blockers may be attainable[67].

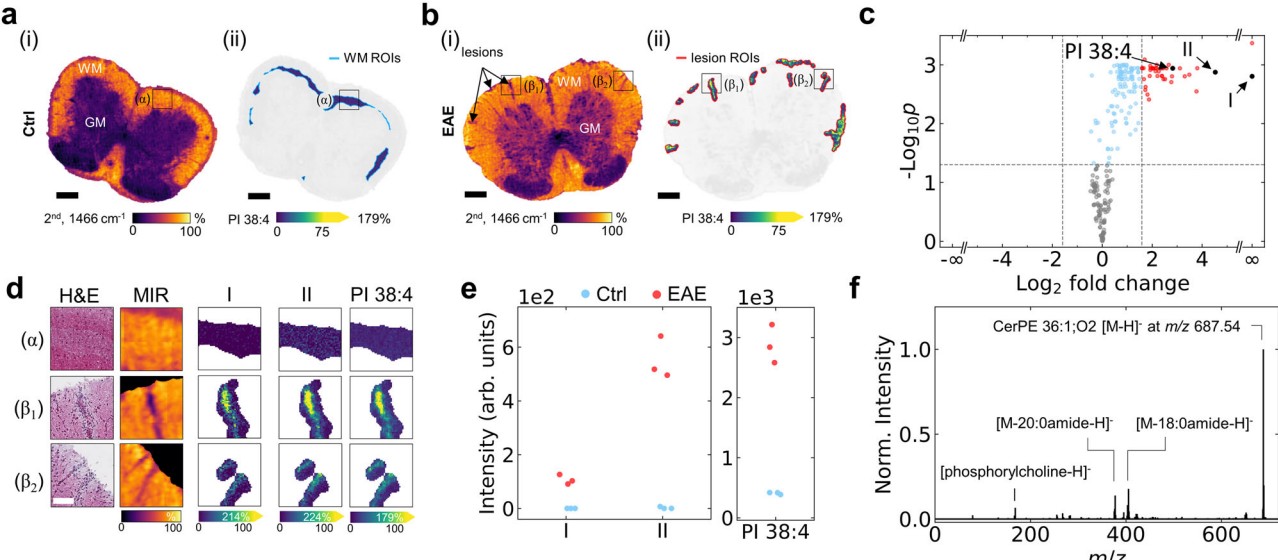

**Fig. 4 | QCL-MIR imaging-guided TIMS-MSI suggests role of proinflammatory lipids in dynamic lipid remodeling in spinal cord white matter of experimental autoimmune encephalomyelitis (EAE). a, b (i)** MIR images of healthy control (Ctrl, **a**) and EAE (**b**) mouse spinal cord at 1466 cm⁻¹ (2nd derivative), with insets (α – no lesion), and (β₁),(β₂ – with lesions) in spinal cord white matter. Scale bar, 300 μm. **a, b (ii)**, QCL-MIR imaging-guided MSI ion images for PI 38:4[M-H]⁻. For comparison, the white matter ROI in control spinal cord (**a**) was computationally shrunk to the size of the total EAE lesions ROI (**b**). **c** Volcano scatter plot reveals significantly enriched *m/z* features (red dots; including **I** CerP 34:1;O2, **II** CerPE 36:1;O2, and PI 38:4 as black dots) in EAE lesions. Horizontal dashed line represents a p-value of

0.05, vertical dashed lines represent a fold-change of 3. Statistical significance was tested by two-sided standard t-test. P-values are Benjamini-Hochberg corrected. **d** QCL-MIR imaging-guided ion images of **I** *m/z* 616.472 (CerP 34:1;O2[M-H]⁻) **II** *m/z* 687.545 (CerPE 36:1;O2[M-H]⁻), and *m/z* 885.549 (PI 38:4[M-H]⁻) for insets (α), (β₁), (β₂) in (**a, b**). MIR and H&E images for reference. Scale bar, 75 μm. **e** Median intensity for **I** CerP 34:1;O2, **II** CerPE 36:1;O2, and PI 38:4 in Ctrl (blue) and EAE (red) spinal cord for n = 3 biological replicates and *n* = 3 technical replicates each. Individual boxplots are depicted in Suppl. Fig. 36. Benjamini-Hochberg corrected p-values were below 0.002 for all three *m/z* features based on a two-sided unpaired t-test. **f** iprm-PASEF MS² analysis of *m/z* 687.545 reveals CerPE 36:1;O2.

In conclusion, with this study we make workflows and computational tools for QCL-MIR imaging-guided high-performance MSI available for the scientific community. This versatile platform is applicable for many biomedical research topics ranging from single-cell metabolomics to 3D cell cultures and to well-defined functional tissue areas in disease research, and it offers enhanced bioanalytical depth while simultaneously reducing measurement time by focusing MSI on relevant ROIs. Consequently, it enables deep spatial profiling of odd-chain lipids, evaluation and improvement of prediction tools or investigation of structure-CCS-relationships. Genetically modified mice can serve as analytical ground truths in MSI, besides their established role as disease models. Altogether, the presented concepts pave the way for deeper spatial investigations of complex biological processes in general and of sulfatide biochemistry in particular with a high level of confidence in molecular identifications.

## Methods

### Ethics statement

Arylsulfatase A-deficiency (ARSA−/−) mouse model: Animal experiments (breeding and maintaining of ARSA−/− mice) were approved by the Landesamt für Natur, Umwelt und Verbraucherschutz Nordrhein-Westfalen (reference: 84-02.04. 2014.A117).

Experimental autoimmune encephalomyelitis (EAE) mouse model: All animal experiments were performed in strict accordance with Austrian law and FELASA guidelines and approved by the Austrian Ministry of Sciences (project number 2022-0.474.463). The study was conducted in accordance with the local legislation and institutional requirements.

### Materials

All chemicals and solvents were of HPLC-MS grade. Conductive indium tin oxide (ITO)-coated glass slides were purchased from Diamond Coatings (West Midlands, UK) or Bruker Daltonics ([ITO slides and

MALDI IntelliSlides], Bremen, Germany). SuperFrost Plus Adhesion slides and BioGold Microarray Slides were obtained from Thermo Fisher Scientific (Schwerte, Germany). MALDI matrix 2,5-dihydroxyacetophenone (DHAP) was purchased from Thermo Fisher Scientific (Waltham, Massachusetts, USA). 1,5-diaminonaphtalene (DAN), α-cyano-4-hydroxycinnamic acid (α-CHCA), 1,2-Di-myristoyl-*sn*-glycero-3-phosphoethanolamine (PE 28:0), and Sulfatides Brain Mix (Avanti Polar Lipids) were purchased from Merck KGaA (Darmstadt, Germany). Acetonitrile (ACN), ethanol (EtOH), LC-MS water, 2-propanol (IPA), ammonium sulfate (AmS) and glass cover slips were obtained from VWR Chemicals (Darmstadt, Germany). Hydroxypropyl methylcellulose: polyvinyl pyrrolidone (HPMC:PVP, 1:1, w/w) was prepared in-house as described elsewhere[68]. For external calibration of the trap unit of the timsTOF fleX and timsTOF Pro mass spectrometer (Bruker Daltonics), ESI-L Low Concentration Tuning Mix (Agilent Technologies, Waldbronn, Germany) was used. Sulfatide standard C₁₇ mono-sulfo galactosyl(β) ceramide d18:1/17:0 (SM4 35:1;O2), and ganglioside standard C18:0 GM3-d₅ were purchased from Avanti Polar Lipids (Birmingham, USA). Trifluoroacetic acid (TFA), Mayer's hemalum solution, hydrochloric acid, sodium bicarbonate, magnesium sulfate, eosin Y-solution 0.5%, xylene, and eukitt were purchased from Merck KGaA. Anti-pan-cytokeratin (panCK) and anti-vimentin antibodies labeled with photo-cleavable mass-tags (PC-MT) were purchased from AmberGen (Billerica, USA).

### Mouse studies

**Arylsulfatase A-deficiency (ARSA−/−) mouse model.** The arylsulfatase A (ARSA) mutant mouse line that was generated using an ARSA gene targeted embryonic stem cell clone[69] has been described previously[32]. Here, mice that had been backcrossed with C57BL/6 J mice for 12 generations were used. ARSA-deficient (ARSA−/−), heterozygous (ARSA + /-) and wild-type (ARSA + /+) mice were obtained from heterozygous breeding pairs. Female mice aged 12 or 60 weeks

were sacrificed by cervical dislocation, organs were removed and immediately frozen on dry ice. Frozen organs were stored at -80 °C and shipped on dry ice.

**Experimental autoimmune encephalomyelitis (EAE) mouse model.**
C57BL/6 J mice (Janvier (#SC-C57J-M)) were bred and kept in a 12 h light cycle, at 21–23 °C and 45-65 % humidity in pathogen-free mouse facilities of the Medical University of Vienna. Experimental autoimmune encephalomyelitis (EAE) was induced in 14 week old male animals by subcutaneous immunization (150 µL per mouse) with 75 µg myelin oligodendrocyte glycoprotein (MOG)$_{35-55}$ (Charité Berlin, Germany) emulsified 1:1 in complete Freud's adjuvant consisting of incomplete Freud's adjuvant (Merck, #F5506-10X10ML) enriched with 10 mg/mL *Mycobacterium tuberculosis* (Difco/BD Pharmingen, #H37Ra). 150 ng Pertussis toxin from *Bordetella pertussis* (Hooke Laboratories, #BT-0105) was administered intraperitoneally at days 0 and 2 post immunization. Spinal cords were harvested from healthy mice (no EAE immunization) or from mice at peak EAE and embedded as previously described[70]. Peak EAE was at days 16–17 post-immunization and defined as animals with total paralysis of both hind limbs, which corresponds to a disease score of three[48].

**3D-cell culture models, tissue and spheroid slice preparation**
Monoculture and biculture spheroids of CCD-1137Sk human fibroblasts and HT-29 human colon cancer cells (both LGC Standards, Wesel, Germany) were prepared, embedded, frozen and cut as described previously[51,68]. Briefly, spheroids were harvested after 3 days and 12 hours after seeding at a density of $1 \times 10^6$ cells/T75 flask for HT-29 cells and $1.5 \times 10^6$ cells/T75 flask for CCD-1137Sk. The spheroid formation was performed with a total number of 10k cells and 20k cells seeded per well for the mono- and bi-culture, respectively, using 96-well cell-repellent microplates (Greiner; cat. No. 650970). This resulted in the formation of a single spheroid per well.

Grown spheroids of the same type were collected in the same Eppendorf tube. Excess culture media was removed and 1 mL of PBS was added for washing. For embedding, spheroids were transferred to HMPC-PVP filled channels inside a gelatin cryo-mold[68]. Fresh-frozen mouse brains, kidneys, spinal cords, and embedded spheroids (-80 °C) were sectioned at 10 and 20 µm thickness, respectively, with a Leica CM1950 cryostat (Leica Biosystems, Nussloch, Germany) at −18 °C chamber- and specimen head temperature. Sections were thaw-mounted onto ITO-coated glass slides from Diamond Coating or IntelliSlides and either stored at −80 °C until further use or dried for a minimum of 15 min in a desiccator. Dried ITO slides were put in a slide mailer and vacuum-sealed to avoid environmental influences on the samples.

**Quantum cascade laser (QCL)-based mid-infrared imaging microscopy**
QCL-MIR imaging was conducted on a Hyperion II ILIM FT-IR and QCL microscope (Bruker Optics, Ettlingen, Germany) equipped with a $300 \times 300$ focal plane array detector and spatial coherence reduction technology[50]. For rapid data acquisition of large specimens, a 3.5x (0.15 numerical aperture (NA)) objective was used, resulting in a nominal pixel size of 4.66 µm. In addition, high resolution images were recorded with either a 15x (0.4 NA) or a 20x (0.6 NA) objective, yielding nominal pixel sizes of 1.15 µm and 0.86 µm, respectively. The optical resolution of the instrument at wavenumber 1500 cm$^{-1}$ can be estimated to be 22.2 µm (0.15 NA), 8.3 µm (0.4 NA) and 5.5 µm (0.6 NA). All measurements were performed in reflection mode using ITO-slides (Supplementary Fig. 2). Prior to data acquisition, a background spectrum was collected on a clean part of the slide. Focus was adjusted manually.

**QCL-MIR imaging in single wavenumber mode.** For image registration and teaching of the mass spectrometer, MIR data was recorded in *single wavenumber mode* at 1656 cm$^{-1}$ (amide I band used for best contrast) for whole slide scans, and images were exported from the OPUS software v8.8 (Bruker Optics) via a python interface as a *.tiff* file.

**QCL-MIR imaging in sweep scan mode for tissue segmentation and definition of regions-of-interest (ROI).** MIR hyperspectral imaging data of tissue specimens was recorded in *sweep scan mode* within a spectral range of 950–1800 cm$^{-1}$, covered by four QCL modules, with a spectral sampling interval of 4 cm$^{-1}$. Individual image tiles consist of $250 \times 250$ pixels. Hyperspectral data cubes were exported from the OPUS software v8.8 (Bruker Optics) via a python interface as *.pickle* file. Hyperspectral data was then imported into an in-house python-based software tool. Subsequently, data pre-processing such as baseline correction and spectral differentiation was performed based on case-specific, user-defined settings. Typically, baseline distortions of absorbance spectra were corrected using asymmetric least square smoothing[12,71] (smoothing factor 1,000,000, weighting factor 0.01, and 10 iterations as default), and piecewise linear interpolation was utilized to calculate the first and second order derivative. For higher order polynomial interpolation and spectral differentiation, a Savitzky-Golay filter is implemented using the *sciPy* package[72]. Other pre-processing algorithms commonly used in mid-infrared spectroscopy like resonant Mie scattering correction[73] could be integrated in future.

For visualization, differentiated spectra were interpolated using a cubic interpolation function. For simplicity, we consider the data obtained from spectral differentiation, usually given in units of 1/cm$^{-1}$ (1$^{st}$ derivative) and or 1/cm$^{-2}$ (2$^{nd}$ derivative), as unit-less.

**Feature-selective image segmentation for definition of regions-of-interest (ROI).** A binary image created by a Gaussian Mixture Model (GGM; two clusters) on the amide peaks usually serves as a mask that distinguishes tissue from background. For feature-selective image segmentation, in step1, use case-specific sets of wavenumbers were defined based on literature or comparison of 2$^{nd}$ derivative data using multimodal approaches. In step2, *k*-means clustering for ROIs definition was utilized on a reduced hyperspectral data cube defined by a selected set of spectral features, if not stated otherwise. Use case 1 – segmentation of brain regions in ARSA−/− mice. For the image segmentation process of distinct brain regions, the lipid-associated spectral features at 1466 cm$^{-1}$ and 1740 cm$^{-1}$ were selected, respectively. Use case 2 – 3D cell biculture models: To distinguish CCD −1137Sk and HT-29 cell lines containing regions in biculture spheroids by image segmentation, spectral bands at 1466 cm$^{-1}$ and 1740 cm$^{-1}$ were selected (Supplementary Fig. 5a). For each monoculture spheroid, an individual measurement area was used. In case multiple spheroids were not spatially separated, individual objects were recognized utilizing *random walker* segmentation during final assignment of MSI measurement regions[74]. Use case 3 – glomeruli in kidney tissue from mice: For identification of glomeruli-containing tissue regions, blob-detection was performed on a mean MIR image created from spectral features of the transmittance data centering around 1726 cm$^{-1}$ and 1142 cm$^{-1}$ (1$^{st}$ derivative of transmittance) (Supplementary Fig. 8c). Initially, the cortex of the kidney was selected by a donut-shaped binary mask preserving the outline of the tissue region. Threshold filtering followed by size exclusion and eccentricity filtering yielded ROIs, the relevance of which was later confirmed by MSI (Supplementary Fig. 8ab). Typically, values for image thresholds were in the range of 30–60 (8-bit image), 40–60 and 1000 pixels for the lower and the upper limits, respectively, of the bandpass filter for size exclusion (3.5x objective, 4.66 µm pixel size) and 0.94 for eccentricity. Use case 4 – neurons in mouse spinal cord tissue: Identification of individual neurons in gray matter of spinal cord samples as in use case 3 with the following modifications: a binary mask of the gray matter region was generated to restrict the blob detection to this specific area of the spinal cord. The 15x objective was selected for image recording.

Parameters for blob detection were adopted accordingly. Use case 5 – kidneys' inner stripe of outer medulla (ISOM) and inner medulla/papilla (IMP) in ARSA−/− mice: Image segmentation was performed using a selected set of characteristic lipid-associated spectral features[75] distinct for the accumulation of sulfatide lipids. The following features were selected: $1466 cm^{-1}$ ($CH_2$ bending vibration), $988 cm^{-1}$ ($C_\beta$-O vibration of the 3-sulfogalactosyl head group), $1740 cm^{-1}$ (C=O stretching vibration) as well as the protein-associated features at $1548 cm^{-1}$ and $1656 cm^{-1}$ (amide II and I bands) (Supplementary Fig. 5b). Subsequently, $k$-means clustering was conducted on the reduced and masked hyperspectral data cube[12]. The number of clusters $k$ was directed by calculation of the Calinski-Harabasz-Score implemented in the *yellowbrick* package[76]. For (semi-)quantitative analysis of lipid accumulation, the MSI signal intensities for 87 sulfatides[32] were summed up, and their distribution within the ISOM defined by MSI image segmentation was deduced for each mouse individually and directly compared with MIR imaging results using a Z-score representation of the data (Suppl. Fig. 14). The distribution of $2^{nd}$ derivative of absorbance values was calculated for distinct spectral features at, e.g., $1466 cm^{-1}$. Use case 6 – white matter and lesions in spinal cord tissue of control and EAE mice: Second derivatives of spectra were calculated using a Savitzky-Golay filter with a third-degree polynomial and a window length of 7. Selected spectral features were $1374 cm^{-1}$ ($CH_3$ symmetric bending vibration), $1466 cm^{-1}$ ($CH_2$ bending vibration), $1548 cm^{-1}$ and $1656 cm^{-1}$ (amide II and I bands) and $1740 cm^{-1}$ (C=O stretching vibration). Hyperspectral data was spatially binned ($2 \times 2$) prior to data import. Selection of white matter lesion ROIs was performed manually after image segmentation (Supplementary Fig. 11).

**Transfer of QCL-MIR imaging-defined ROIs to the data acquisition file of the mass spectrometer and teaching.** MSI data acquisition was initially set up in flexImaging v7.2/v7.4 (Bruker Daltonics), while using the whole-slide single $1656 cm^{-1}$ wavenumber reference image (transmittance data) to teach the MS device. Hereby, the single-wavenumber image is modified by affine transformation and further used to generate a data acquisition file (.*mis*). The corresponding image at $1656 cm^{-1}$ from the hyperspectral cube is co-registered with the modified reference image by means of *SimpleITK*[77] and *simple Elastix*[78] yielding an affine transformation matrix, which is further used to transfer the ROI information into the frame of the modified image. Advanced Mattes mutual information was used as a similarity metric using linear interpolation. QCL-MIR imaging-defined ROIs were further processed in Python v3.8 to account for use case-specific demands, e.g. by removal of small holes (all use cases), size reduction (spheroids, ISOM, IMP) or expansion (glomeruli and neuron) by a layer of one or two MALDI pixels using respective functions from the *scikit-image* and by an in-house written iterative approach for hole opening to account for donut-shaped measurement regions (ISOM and IMP)[79]. For white matter regions of spinal cord, an erosion operation was applied with an erosion factor of 25 on a 4.66 μm pixel-sized single-wavenumber image. In contrast, neuron-containing regions were expanded by a factor of 4. Finally, ROIs were imported as polygonal areas into the .*mis* file.

**Image co-registration for multimodal imaging.** Image co-registration for multimodal data analysis was performed as previously described in ref. 80. To estimate performance of segmentation/identification of small objects like glomeruli-containing tissue areas, we compared the number of identified objects in a multimodal approach, *i.e.* by calculation of the number of objects in the intersection of both modalities (MIR ∩ MSI) relative to the number of objects unique to each modality (Supplementary Fig. 9). Parameters for blob-detection were optimized to yield a high ratio between the numbers of objects identified in both modalities vs. MIR alone. For identification of glomerular regions in MSI, we applied our previously described concept for spatial probabilistic mapping of metabolites[2] to generate glomerular hotspots. For visualization purposes and incorporation of ROIs SCiLS Lab (Version 2024a Pro, Bruker Daltonics) and the SCiLS Lab API v6.3.115 has been used.

To tissue prevent degradation due to environmental conditions, the slides were either processed immediately or carefully placed in a slide mailer, vacuum-sealed and stored at −80 °C for optimal sample preservation.

**MALDI-TIMS-Mass Spectrometry Imaging**
**Calibration.** Prior to MSI data acquisition, external mass and ion mobility calibration was achieved (via ESI source) using ESI-Low Concentration Tuning Mix (Agilent Technologies, Santa Clara, USA) and a linear calibration model. Final mass calibration via the MALDI source was performed using red phosphorus (RedP) clusters $P_n$ ($n = 13$–61 in intervals of 4) and an enhanced-quadratic calibration model. During data acquisition, the internal standard (IS) SM4 35:1;O2 ($C_{41}H_{79}NO_{11}S$, [M-H]−; m/z 792.530107) was used for internal lock-mass calibration.

**MALDI trapped ion mobility spectrometry MSI.** MALDI TIMS-MSI was carried out on a timsTOF fleX system (Bruker Daltonics) equipped with a smartbeam 3D 10 kHz laser and TimsControl 4.1/5.1 and flexImaging v7.2/v7.4 software (Bruker Daltonics). Data was acquired in negative ion mode with 240 laser shots per pixel, 2 kHz laser frequency and lateral step size 20 μm. For the m/z range of 600–1200, the Ion Transfer parameters were as follows: MALDI Plate Offset 50 V, Deflection 1 Delta −70 V, Funnel 1 RF 350 Vpp, isCID Energy -0.0 V, Funnel 2 RF 350 Vpp, and Multipole RF 320 Vpp. Collision Cell parameters: Collision Energy 8 eV, and Collision RF 1600 Vpp. Quadrupole parameters: Ion Energy 5 eV, and Low Mass m/z 320. Focus Pre TOF parameters: Transfer Time 85 μs, and Pre Pulse Storage 10 μs. For acquisition of full kidney datasets, the qTOF mode was utilized, and laser frequency was 10 kHz. TIMS-ON-MSI data was acquired in a range of 0.80–1.87 Vs/cm², with ramp time 480 ms, and accumulation time 120 ms (resulting duty cycle 25 %). Tims-offsets: Δt1 (Deflection Transfer -> Capillary Exit): 20 V; Δt2 (Deflection Transfer -> Deflection Discard): 120 V; Δt3 (Funnel 1 In -> Deflection Transfer): -80 V; Δt4 (Accumulation Trap -> Funnel 1 in): -100 V; Δt5 (Accumulation Exit -> Accumulation Transfer): 0.0 V; Δt6 (Ramp Start -> Accumulation Exit): -100 V; Collision Cell In: -225 V. TIMS-ON-MSI data of glomeruli was acquired at a lateral step size of 10 μm. Transfer time and pre-pulse storage were increased to 120 μs and 15 μs, respectively. The range was adjusted to 1.20–2.05 Vs/cm², and offsets were as follows: Δt1 (Deflection Transfer -> Capillary Exit): 20.0 V; Δt2 (Deflection Transfer -> Deflection Discard): 120 V; Δt3 (Funnel 1 In -> Deflection Transfer): -85 V; Δt4 (Accumulation Trap -> Funnel 1 in): -150 V; Δt5 (Accumulation Exit -> Accumulation Transfer): 0 V; Δt6 (Ramp Start -> Accumulation Exit): -150 V; Collision Cell In: 225 V. TIMS In gas flow was 2.680 mbar. For the m/z range of 1200–2000, Funnel RF 1 and Funnel RF 2 were adjusted to 500 Vpp, Multipole RF to 600 Vpp, Collision Energy to 10 eV, Collision RF to 2500 Vpp, Low Mass to m/z 620, Transfer Time to 150 μs, Pre Pulse Storage to 15 μs, and Collision Cell In to 300 Vpp. The mobility range was 1.20–2.30 Vs/cm². TIMS In gas flow was 2.100 mbar. Acquired CCS values were recalibrated based on a subset of SM3 acquired with 2.680 mbar gas flow.

**On-tissue lipid/metabolite fragmentation using iprm-PASEF.** Parallel reaction monitoring with parallel accumulation and serial fragmentation (prm-PASEF) of lipids/metabolites has so far mainly been outlined for TIMS-MS without spatial resolution[55], for LDI-MSI of tattoo pigments[14] and for on-tissue MALDI-MSI fragmentation of bespoke biomolecules[15]. Here, we used a prototype version of Bruker software for spatially resolved imaging prm-PASEF (iprm-PASEF) data acquisition in the m/z range 50–2000 with a quadrupole low mass of m/z 50, a collision cell energy of 4 eV and collision RF of 500 Vpp. The laser

frequency was 5 kHz, the transfer and the pre-pulse storage times were 65 μs and 8 μs, respectively for kidney ISOM and IMP, and 70 μs and 10 μs, respectively, for glomeruli. Fragmentation energies were modulated for kidney ISOM and IMP (2.680 mbar: for $1/k_0 = 1.25$ Vs/cm$^2$, collision energy 65 eV; 1.30 and 72; 1.35 and 80; 1.40 and 85; 1.45 and 95; 1.60 and 100. 2.100 mbar: 1.50–1.90 and 100; 2.00 and 85, 2.10 and 80, 2.30 and 70), for glomeruli (1.20 and 55; 1.70 and 75; 1.80 and 80), and for motor neurons and lesions in mouse spinal cord (1.25 and 40; 1.30 and 40; 1.35 and 40; 1.40 and 50; 1.45 and 55; 1.60 and 60). For kidney samples, all iprm-PASEF data was acquired at lateral step size 40 μm leading to a total number of <1500 pixels (<10 min acquisition time) for each IMP and ISOM ROI respectively. For each iprm-PASEF acquisition, up to a maximum of 15 precursors were analyzed in parallel and overlaps in their particular mobility traces were ruled out.

Multiplex-MALDI-MS-Immunohistochemistry (IHC) using two PC-MT antibody probes (pan-cytokeratin (pan-CK) and vimentin) was carried out as described in ref. [52], with the following modifications for fresh- frozen spheroid sections: 2 × 3 min ice-cold acetone, 30 min 1% paraformaldehyde fixation; 10 min PBS; 2×3 min acetone; 3 min Carnoy's solution; 2 × 2 min 100% ethanol, 3 min 95% ethanol, 3 min 70% ethanol, 3 min 50% ethanol and 10 min TBS. Antigen retrieval (100x Tris-EDTA Buffer, pH 9) was performed for 30 min using a water bath at 95 °C in a coplin jar (VWR Chemicals). Spheroid sections were blocked with Tissue Blocking Buffer (2% (v/v) normal mouse serum and 5% (w/v) BSA in TBS-octyl-beta-glucoside (OBG; 0.05% (w/v)). For the PC-MT antibody treatment, the slide was incubated with 3 μg/mL of pan-CK and 2 μg/mL of vimentin each antibody diluted in Tissue Blocking Buffer at 4 °C overnight in a humidified, light-protected environment. Next, the slide was washed with 3 × 5 min TBS; 3 × 2 min 50 mM ammonium bicarbonate, and dried in a vacuum desiccator for 2h. Probes were photo-cleaved at 365 nm for 10 min in a UV illumination box (AmberGen) and α-CHCA MALDI matrix was applied. Data was acquired using a timsTOF fleX mass spectrometer. PC-MTs were analyzed in qTOF mode with the following ion transfer parameters: MALDI Plate Offset 30 V, Deflection 1 Delta 80 V, Funnel 1 RF 500 Vpp, isCID Energy ZA0.0 V, Funnel 2 RF 500 Vpp, and Multipole RF 1200 Vpp. The Collision Cell parameters were: Collision Energy 25 eV, and Collision RF 4000 Vpp. The quadrupole parameters were: Ion Energy 15 eV, and Low Mass m/z 900. The Focus Pre TOF parameters were: Transfer Time 140 μs, and Pre Pulse Storage 10 μs.

### Analysis of MALDI-MS Imaging data

**Selection of hypothetical sulfatide configurations.** Based on previous empirical data[32], we considered the fatty acid compositions from 32:x to 46:x, with x = 0,1,2,3 for O2,O3 and x = 0,1 for O4, and six lyso-sulfatide compositions as potentially detectable with our MSI settings, resulting in a total number of 156 theoretical compositions for each sulfatide subclass.

**Classification of sulfatide peaks in qTOF mode.** Each sulfatide peak that showed at least one additional peak within a mass window of ±1.1 Da that exceeded the intensity of the first sulfatide isotope peak was classified as "non-clean", since this "interfering" peak would prevent an unequivocal assignment of an on-tissue MS$^2$ spectrum (Supplementary Fig. 18c). Sulfatide peaks without interfering peaks were classified as "clean" (Supplementary Fig. 18a). *Classification of sulfatide peaks in TIMS-ON mode:* Each sulfatide peak that shows a colocalizing peak within a mass window of ±1.1 Da and a mobility window of ±0.005 Vs/cm$^2$ that exceeds the intensity of the first sulfatide isotope peak was classified as non-clean, since this interfering peak would prevent a clear assignment of an iprm-PASEF spectrum. Sulfatide peaks without interfering peaks were classified as clean. *TIMS-MSI:* CCS values obtained from QCL-MIR imaging-guided MSI of four biological replicates were averaged and the standard deviation was calculated and given as uncertainty. To obtain CCS values from prediction tools

AllCCS2[34], DeepCCS[36] and LipidCCS[61], simplified molecular input line entry specification (SMILES) strings were provided to these tools. All chemical structures and all SMILES strings were generated in Chem-Draw 21.0.0.38 (PerkinElmer, Waltham, US). To benchmark the predicted values CCS$^{pred.}$ against experimental values CCS$^{MSI}$, the mean relative deviation $\bar{\varepsilon}$ in % was calculated by

$$\bar{\varepsilon} = 100 \cdot \frac{\left| CCS^{MSI} - CCS^{pred.} \right|}{CCS^{MSI}}.$$

For the evaluation of relative differences in CCS values, e.g., of the degree of glycosylation in the sulfated head group and the α-hydroxylation of the N-acyl-linked FA in sulfo-glycosphingolipids, a global least square model fitting was applied. Hereby, a 2$^{nd}$ order polynomial fit was performed by keeping the amplitudes of the linear and quadratic term fixed, parallel line model, when describing the data for two (or three) similar sulfatide subclasses, e.g., SM3 18+n:1;O2 and SM4 18+n:1;O2. On the level of our experimental accuracy, we found that this parallel line model can be applied to describe the data for similar subgroups, enabling us to deduce relative contributions of chemical modifications to CCS values.

### LC-TIMS-TOF-MS

**Sulfatides extraction from ARSA−/− kidney tissues.** Two 12-week-old and two 60-week-old ARSA−/−, two heterozygous (ARSA+/-) and two wild-type (ARSA+/+) kidney samples underwent two rounds of homogenization for 30 seconds each using 400 μL of 200 mM Na$_2$CO$_3$ (pH=9.3). Subsequently, all samples were subjected to overnight freeze-drying at -56 °C and 1 mbar.

The extraction of kidney tissues was carried out utilizing a modified Folch method. Kidney tissues, spiked with 5 μL GM3-d5 (100 μg/ml in methanol) and 5 μL SM4 35:1;O2 (100 μg/mL in chloroform/methanol, 2:1[v/v]) as internal standards, were mixed and sonicated with 9 mL of chloroform/methanol (2:1[v/v]) for 15 min. Subsequently, 840 μL of water was added, and the resulting mixture was thoroughly mixed before being centrifuged at 3000 rpm for 5 min at RT. The chloroform layer at the bottom was collected and evaporated using a stream of nitrogen. Finally, the samples were re-dissolved in 200 μL methanol/water (4:1[v/v]).

**LC-TIMS-MS.** 4D LC-TIMS-MS experiments were conducted using timsTOF PRO instrument interfaced with an Elute UHPLC system (both Bruker Daltonics). Analysis of sulfatides was performed in negative ion mode. The LC-TIMS-MS method was adapted from the protocol of Lerner et al. (2023)[56] with modifications. The LC column was a C$_{18}$ Kinetex column (100 × 2.1 mm × 2.6 μm) (Phenomenex, Germany). The PASEF scan mode was performed on a mass scan range of m/z 200–2000 Da for both MS and MS$^2$ acquisition. The collision energy was 90 eV.

### Statistics and reproducibility

Experiment visualized in Fig. 1e−g were conducted for each biological replicate as demonstrated in Supplementary Figs. 9, 11 and 13. Experiments presented in Fig. 4a, b, d were conducted for n = 3 biological replicates and n = 3 technical replicates as highlighted in Supplementary Fig. 34.

### Reporting summary

Further information on research design is available in the Nature Portfolio Reporting Summary linked to this article.

## Data availability

Extensive data (4D-LC-TIMS-MS; TIMS-MSI data including fragment elucidation and images) is provided as Supplementary Data. MALDI MSI data is available through the Metaspace portal (https://

metaspace2020.org/project/gruber-2025). MIR imaging data (raw data files) and MSI data (iprm-PASEF data files) acquired in this study have been deposited in Zenodo under accession code https://zenodo.org/uploads/15209646. Processed data are available upon request from the corresponding author C.H. Source data are provided with this paper.

## Code availability

The source code and an executable file of the underlying software tool is publicly available on GitHub. https://github.com/CeMOS-Mannheim/QCL_MIR_guided_MSI.

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

## Acknowledgements

We thank Lucas Gast for performing automated H&E staining, Marten Seeba, Domenic Dreisbach and Niels Kröger-Lui (Bruker Optics) for introduction to the Hyperion II ILIM instrument and computational support, Ethan Yang (Bruker Daltonics) for fruitful discussions, Hans-Christian Koch (Bruker Optics) for access to the Hyperion II instrument, and Arne Fütterer (Bruker Daltonics) for access to iprm-PASEF prototype software. C.H. acknowledges support by the Ministerium für Wissenschaft, Forschung & Kunst (MWK) Baden-Württemberg „Mittelbauprogramm". This work was supported by the BMBF (German Federal Ministry of Research) as part of the Innovation Partnership "Multimodal Analytics and Intelligent Sensorics for the Health Industry" (M²Aind), projects "Drugs4Future" (grant 12FH8I05IA) and "DrugsData" (grant 13FH8I09IA) to C.H. and R.R., within the framework FH-Impuls. The study was also supported by Deutsche Forschungsgemeinschaft (DFG), project "lipid imaging and lipidomics" (No. 511488495) within SFB 1638 to C.H. The financial support to O.S. and G.S. by the Austrian Federal Ministry for Labour and Economy and the National Foundation for Research, Technology and Development and the Christian Doppler Research Association is gratefully acknowledged. Acquisition of the timsTOF fleX mass spectrometer was supported by BMBF as part of the MSCorSys SMART-CARE (grant 161L0212F) to C.H.

## Author contributions

L.G. and S.S. designed and conducted all QCL-MIR imaging experiments, designed all MSI experiments, performed ion mobility analysis and generated the Figures. S.S. wrote python code for multimodal image registration and for MIR data analysis and analyzed MIR data. L.G. conducted all MSI experiments, analyzed the data and interpreted all MSI fragmentation spectra for ground truth evaluations. T.E. wrote R code and performed MSI feature identification. D.A.S. conducted generation of molecular probabilistic maps for multimodal analysis and benchmark of glomerular structures. J.L.C. performed statistical analysis and provided input for ion mobility analysis. T.B. performed EAE spinal cord experiments and analyzed data. M.K., O.S. and G.S. genotyped and provided EAE mouse spinal cords. H.G.V. and L.B. conducted and analyzed all LC-TIMS-MS experiments and generated corresponding tables. F.K. and R.R. performed 3D cell culture experiments and analyzed data. S.A.I. performed MSI 3D cell culture experiments and analyzed data. Y.U. performed multiplex-MALDI-IHC experiments and analyzed corresponding MSI data. O.S., G.S., R.R., L.B. and C.H. provided infrastructure. M.E. genotyped and provided ARSA–/– mouse organs

and insights into sulfatide biochemistry. C.H. conceived and managed the overall study and wrote the first draft of the manuscript. L.G., S.S. and C.H. wrote the final manuscript – with input from all co-authors.

## Funding

## Competing interests

Bruker Daltonics co-funded the BMBF-funded projects "Drugs4Future" and "DrugsData" within the framework M²Aind, as mandated by BMBF. All other authors declare no competing interests.

## Additional information

¹Center for Mass Spectrometry and Optical Spectroscopy (CeMOS), Technische Hochschule Mannheim, Mannheim, Germany. ²Medical Faculty, Heidelberg University, Heidelberg, Germany. ³Clinical Lipidomics Unit, Institute of Physiological Chemistry, University Medical Center, Mainz University, Mainz, Germany. ⁴Mannheim Center for Translational Neuroscience (MCTN), Medical Faculty Mannheim, Heidelberg University, Mannheim, Germany. ⁵Institute for Vascular Biology, Centre for Physiology and Pharmacology, Medical University of Vienna, Vienna, Austria. ⁶Christian Doppler Laboratory for Arginine Metabolism in Rheumatoid Arthritis and Multiple, Vienna, Austria. ⁷Christian Doppler Laboratory for Immunometabolism and Systems Biology of Obesity-Related Diseases (InSpiReD), Vienna, Austria. ⁸Institute of Biochemistry and Molecular Biology, University of Bonn, Bonn, Germany. ⁹These authors contributed equally: Lars Gruber, Stefan Schmidt. ✉e-mail: c.hopf@hs-mannheim.de

