## [Transparent Peer Review file · Nature Communications]

Deep MALDI-MS Spatial Omics guided by Quantum Cascade Laser Mid-infrared Imaging Microscopy

Corresponding Author: Professor Carsten Hopf

Version 0:

Reviewer comments:

Reviewer #1

(Remarks to the Author)

The manuscript "Deep MALDI-MS Spatial 'Omics guided by Quantum Cascade Laser Mid-infrared Imaging Microscopy" by Gruber et al., discusses the combination of infrared imaging and MALDI for spatial 'OMICS. The combination is interesting. While there has been some work combining infrared imaging based on FTIR microscopy and MS, I don't know many prior publications using EC-QCLs for imaging in combination with MS but now that Bruker (a co-funder of this work) also sells QCL microscopes, we can likely expect a more concerted push towards popularizing the combination. Unfortunately, the work has some significant issues and – in my opinion – should not be published without significant revisions.

Above all, it appears, the authors main concern lies on demonstrating that they did everything there is to do in coupling mid-IR laser spectroscopy and MALDI. The manuscript wants to do too many things at once and thus cannot spend sufficient time on any single aspect of the research to be worthwhile to the reader. English expression and grammar is weak and many nonsensical sentences remain. Figures appear to be present just to fill space or provide eye candy without conveying meaningful information. In some places the manuscript is unintentionally(?) funny. For example, the phrase "at very mass resolution" (pg. 10, ln. 274) would be more more at home on a picture of a Shiba Inu than in a scientific manuscript.

Overall, the mid-infrared spectroscopy part of this manuscript is weak. For example, pg 22 ln 606 claims that "asymmetric least square[sic] smoothing [was used to] correct baseline distortion and light-scattering effects" Leaving aside that it might be more fitting to cite Eiler's original work rather than one of the authors' previous papers, asymmetric least-squares based baseline correction cannot correct light-scattering effects. A method like Bassan's RMieS-EMSC would be required to do this, asymmetric least-square only corrects the baseline offset, one of the effects of scattering. In general, spectral preprocessing is not explained at the depth that would be required. For example, the authors calculate second derivatives of spectra but do not explain how this derivative is carried out. Was Savitzky-Golay used? Which parameters?

Sentences like "QCL-IRI spectral features discriminating between tissues, tissue morphologies, or cell types are then extracted from the IR "fingerprint" spectra and used to segment QCL-IRI data, e.g. by performing 2nd derivative spectroscopy (Fig. 1a(ii)), and to define ROIs (Fig. 1511a(iii))." appear throughout the work. Sentences, that have obvious grammatical issues but are also factually incorrect: calculating the second derivative is not an algorithm used for segmentation but for preprocessing mid-IR spectra.

The work introduces acronyms like "QCL-IRI" that are not used in the mid-IR community but only in this manuscript. The figures depict infrared data in way that makes it essentially useless (no units on axes, interpolation without showing actually measured points). In Fig. 1 a (iii), a(vi), f,g, i, Fig. 2b, Fig. 3C images have no color bar and the depicted information is thus meaningless. The same is repeated for figures in the supplementary. The color scheme in Fig. 2c is puzzling. According to figure caption and legend three groups (I: orange, II:red, III:green) are depicted. However, the image itself does not contain solid colors but gradients. It is not clear that the gradient colors mean or, for some of the hues, which of the three groups they should be assigned to.

In closing, this work obviously is based on a tremendous amount of challenging work. I believe that after fixing the unfortunate factual and textual issues, it will be great contribution to an emerging field.

(Remarks on code availability)

The authors promise to share code in the abstract but I was not able to find any code with the provided materials.

Reviewer #2

(Remarks to the Author)

In this manuscript, Gruber et al. employed QCL-IRI-guided MSI technology to confine highly interested areas for spatial lipidomics profiling at maximized analytical depth with MR-MS or PRM-PASEF-MS analysis. The authors demonstrated the applications of the technology in various types of samples such as cells and tissues, illustrating the advantages of the technique in deep lipidomics profiling. However, the uses of different imaging modalities for selective MSI sampling have been already described by several publications, making the manuscript less innovative (e.g., <https://doi.org/10.1021/jasms.0c00128>, <https://doi.org/10.1021/acs.analchem.5b03610>, <https://doi.org/10.1021/acs.analchem.8b02885>, <https://doi.org/10.1039/C5SC03782D>). Moreover, the IRI-guided MSI strategies have been also previously published according to the authors (<https://doi.org/10.1038/s41598-017-18477-6>), only the tool used to find ROIs was changed.

Most importantly, MSI technology is developed for comprehensive spatial-omics profiling. MSI technology is featured to explore molecular features that other technologies such as immunohistochemistry and IRI cannot differentiate. Thus, relying on ROIs defined by IRI for MSI-based spatial profiling would not support novel biological discoveries. Additionally, the accuracy of lipid identification needs further improvement in this work. More statistical data should be provided to demonstrate the advantages of the strategy used in this work.

Finally, the following major concerns should be addressed before further consideration.

Major comments:

1. The procedure for lipid identification was unclear. If I understand correctly, for data acquired by TIMS-MSI based prM-PASEF fragmentation, only exact mass match was applied for annotation of sulfatide in ARSA -/- mouse kidney (Table 1). To achieve deep lipidomics profiling, it is better to use multi-dimensional information such as CCS and MS/MS match for high-confident lipid identification. For data acquired by MR-MSI, exact mass match and IFS analysis improved identification confidence. However, only 34 and 39 out of 91 and 97 spectra were supported by IFS information. How does the author interpret the annotations that do not conform to IFS analysis? I want to know whether the isobaric overlappings among different lipid species interfered IFS analysis, such as [M+2] isotopologue of SM4 32:2;O2 and M0 of SM4 32:1;O2. For data acquired by LC-TIMS-MS, 4D-descriptor match was applied for lipid annotation as described in Methods. It is better to provide more details for the establishment of 4D-descriptors.
2. Page 6 line 169 & Page 8 line 221: The authors used sets of wavenumbers to define ROIs and validated the feasibility of the QCL-IRI with other techniques such as multiplex-MALDI-IHC and MSI. I would like to know if there are guidelines for choosing case-specific sets of wavenumbers in QCL-IRI analysis for various types of samples. How to ensure that the chosen areas are indeed of interest? If additional methods are required for verification each time, the applicability of the technology may be compromised. Additionally, these applications relied on prior knowledge, suggesting that the technology may have limitation for new biological discovery. Defining ROIs with IRI analysis, which provides limited information, for further MSI analysis may result in the loss of valuable information. The authors should provide new demonstration examples to support new biological discovery instead of validation with known examples.
3. Page 10 line 271: In this part, the authors combined QCL-IRI with MR-MSI for confident lipid identification. In Figure 2, the authors emphasized the advantages of ultra-high resolution MR-MSI. But the assistance provided by QCL-IRI was not adequately explained in the main figure. It is not novel to use ultra-high resolution mass spectrometry and FDR-controlled analysis for lipid identification (<https://doi.org/10.1126/science.abh1623>).
4. Page 14 line 373: The authors focused to the ISOM and IMP ROIS using TIMS-MSI with 2D-mobilogram-based prM-PASEF, which provided additional dimensions for lipid identification. More detailed description of the prM-PASEF method is needed (e.g., how many MS2 spectra were collected for each pixel; how was the prM-list generated). In addition, were product ions used for quantification in prM-PASEF method? Additionally, evaluations of the quantitative accuracy and reproducibility are needed.
5. Page 14 Figure 3: The authors illustrated the advantages of TIMS separation in QCL-IRI-guided MSI via several examples. Several works have already demonstrated the advancement of TIMS in MSI field (<https://doi.org/10.1038/s41467-023-40512-6>, <https://doi.org/10.1038/s41467-023-43298-9>). It is necessary to highlight the combination of QCL-IRI with MSI analysis rather than simply illustrating the advantages of TIMS separation through examples. Additionally, supplementing statistical data rather than examples would enhance the reliability of conclusions.
6. Page 16 line 447: Structure-CCS relationships of experimentally annotated sulfatide lipids were explored. Further, predicted CCS values were summarized via experimental lipidomics data. As a resource, the information accuracy is very important. As mentioned in the 1st comment, it is necessary to provide accurate lipid identifications at first.

Other minor comments:

1. Page 12 line 317: Please provide details of cross-modality (semi-)quantitative analysis in Methods.

2. The figures lacked essential labels, such as those in Figure 1c, Figure 1f, Figure 2i, and so on.
3. It is advised to carefully review the manuscript as there are several typos present. There is a lack of data in Supplementary Figure 22.
4. Please check the citation of the references. There might be an error in the citation of reference 48

(Remarks on code availability)

Reviewer #3

(Remarks to the Author)

(Remarks on code availability)

Version 1:

Reviewer comments:

Reviewer #1

(Remarks to the Author)

I am happy to see that the authors have further improved the manuscript. While there may still be some points that seasoned IR experts might quibble about, this is one of the first papers in an emerging field and no doubt follow up work will be required to explore details.

For future work, I can not emphasize enough the importance of considering standing wave effects and scattering effects in infrared spectral imaging. The field has painfully learned the strong effects scattering has on spectra and that it not only affects the baseline but also relative intensities in the late 2000s. Open software packages for correction exist. e.g. in the OpenVibSpec code. I also want to note that all of these effects are seen especially in transfection mode (e.g. when using ITO covered slides).

(Remarks on code availability)

I am not sure if the code was not provided with the previous submission or if I missed it. This submission includes code. I have verified that it looks like reasonable code for this tasks but I haven't executed it or tested it.

Reviewer #2

(Remarks to the Author)

I regarded this manuscript as a methodology paper aiming at spatial lipidomics profiling at maximized analytical depth with QCL-MIR imaging-guided TIMS-MSI prm-PASEF analysis. After reading the revised manuscript and the detailed workflow, I kept my concerns about this technology. I would not recommend it for publication in Nature Communications.

The major concerns in the previous comments were NOT fully addressed (Comments #1, #2, #4, #5 and #6, and the major concern). I will not take the time to repeat these concerns.

To illustrate with a simple example: the study includes enough technical replicates but lacks biological replicates. I highlighted this issue in my previous comments, yet it was entirely ignored by the authors. This disregard makes me feel that reviewing this work again is a waste of time and reflects a lack of respect, transparency and honest. Some other concerns were addressed in the similar way. I will not repeat again.

In addition, there are additional concerns for the revised manuscript:

1. The authors mentioned that "Parallel reaction monitoring with parallel accumulation and serial fragmentation (prm-PASEF) of lipids/metabolites has so far only been outlined for TIMS-MS without spatial resolution." In fact, similar spatially resolved analysis has already been conducted in previous work (<https://doi.org/10.1038/s41467-023-43298-9>). Therefore, the novelty of prm-PASEF for tissue-based MS/MS is diminished.

2. MSI technology is developed for comprehensive spatial-omics profiling. Relying on ROIs defined by IRI for MSI-based spatial profiling would result in the loss of some potential biological information, making it insufficient to support new discoveries. The results shown in Supplementary Fig. 8 indicated that the MSI-unique regions were more extensive than the regions shared by both MSI and MRI. If the ROI was defined based on MRI results, the biological information in these MSI-unique regions would have been overlooked.

3. A more detailed description of lipid identification is provided, but some issues remain. I just pointed out some examples. There are many others, and I will not take my time to repeat these concerns.

1) For lipids identified using prm-PASEF analysis in Supplementary Table 7, the authors choose M3 of PI 34:1 ([M-H]⁻). Why choose the M3 instead of M0?

2) PI 36:3 and PI 18:0/20:3 were NOT isomeric structures. In fact, their mass differs by 28 Da. However, in Supplementary Table 8, they were annotated in a chimeric spectrum.

3) The authors set the isolation window as ± 1.1 Da. Why did the MS/MS spectrum annotated as PI 34:1 showed interference from PI 40:3 fragments?

Additionally, in Line 360, the authors suggested that lipids detected only in TIMS-MSI were due to the different mass and mobility ranges in LC-TIMS-MS. It is better to use the same mass and mobility ranges in LC-TIMS-MS analysis for comparison. With the help of LC separation, it could have assisted in confirming whether SM1 a/b originated from the in-source fragmentation of SB1a.

4. When comparing lipid changes between EAE mice and control mice, individual differences were not considered, and no biological replicates were included. It would be better to characterize the lipid remodeling in spinal cord with the lesions and their adjacent normal areas from the same mouse.

Overall, I kept my concerns about the novelty and essentiality of the developed method. I am skeptical about the accuracy of the lipid identification. The validations of quantitative accuracy and reproducibility are needed. The application study lacks of biological replication. And many others!

(Remarks on code availability)

Reviewer #3

(Remarks to the Author)

(Remarks on code availability)

Version 2:

Reviewer comments:

Reviewer #4

(Remarks to the Author)

The manuscript by Gruber et al details the introduction and application of quantum cascade laser mid-infrared imaging microscopy (QCL-MIR imaging). The authors developed QCL-MIR imaging-guided MSI workflows and computational tools for spatially resolved deep lipidomics profiling and made them and extensive spatial lipidomics data available as a community resource. QCL-MIR imaging-guided spatially resolved on-tissue MS² of lipids by iprm-PASEF using ion mobilograms was performed and benchmarked using a ground truth animal model where specific lipids, sulfatides, were known to be dysregulated and previously characterized. Overall this is a very well done study I have tried to parse out my own critiques and comments with also being asked to evaluate prior reviewer comments without being an original reviewer.

Major

1. While the authors do address adding the 2023 Nat. Commun SIMSEF paper, a statement remains in the abstract that should be altered to match the updated introduction, that current MSI only offers "putative" molecular identifications" is incorrect a number of instruments are capable of acquiring MS/MS and ion mobility CCS values. This sentence should be adjusted in light of the SIMSEF paper that is later mentioned and discussed. I do agree that most people still don't use this but I believe this is more of a data size and handling issue rather than not being possible on a number of different instruments.

2. "Dedicated computational tools for information-rich and high-throughput QCL-MIR imaging-guided MSI are lacking" The experimental data and the code should be made publicly available reported in this manuscript. A major promise and premise of this workflow and the novelty lies in the integration of the data as outlined in SI Figure 1. It is unclear how exactly the data analysis proceeds and the timeline for the analysis. As it stands, it is unclear exactly how one would replicate these results from the IT workflow standpoint, which is where the major novelty of this manuscript lies. Please make this and the data publicly available. This would seem like a major impediment to wide adoption of this technology to not readily release the code needed.

3. In SI Figure 1, do parts 3 and 4 segmentation and ROI transfer occur offline from both instruments? Can the authors comment on the time for each of these steps to give a feel for how long this might take and storage recommendations for the

tissues during the offline analysis?

Minor

1. Regarding SI Figure 5, could the authors comment on the error associated with the QCL-MIR measurements. These differences they highlight seem very minor or within the margin of error in other types of measurements. I am not expressly familiar with this measurement so providing some level of acceptable interpretation and error would be helpful for a generalist reader.
2. On a couple of occasions the authors refer to “computationally dilated” do they mean enlarged, the meaning of this is unclear.

Comments

1. I appreciated the including of data in SI Figures 3-4 this was nicely done and highlights that the QCL-MIR is non destructive to the sample and the lipids being queried.
2. The amount of data provided in the SI provides a high level of rigor to this manuscript. The inclusion of the different controls with replicates and the MS/MS spectra is well done including key experimental details such as the CE.
3. The data comparisons and increased identifications presented in the section entitled “QCL-MIR imaging-guided trapped TIMS-MSI enables extensive fragmentation analysis on tissue and sulfatide identifications on par with LC-based 4D-TIMS-PASEF” highlights the utility of this method, although it was also nice to see that the identification numbers weren’t vastly different but enough to highlight that targeted methods for directed analysis are beneficial especially for a more comprehensive analysis than is currently being conducted during most general experiments. This strikes a nice balance for utility and need.

Comments regarding other reviewer’s concerns

1. As a new reviewer of this manuscript, regarding 2.4.2 Concern 2, the authors do an excellent job of highlighting the tradeoffs between a fully untargeted 4D MALDI-TIMS-qTOF experiment and time and using structural features provided by MIR to guide a targeted in-depth analysis. At no point do they propose that this methodology should replace what labs are currently doing, but rather this new methodological workflow provides a different, orthogonal route for probing biomedically relevant questions.
2. The lipid identifications are well done and sufficient data including MS/MS and CCS values in relevant models in which these molecules are known to be implicated is provided. I have no concerns regarding the assignment of the lipid species in this manuscript. The SI was expensive and comprehensive.

(Remarks on code availability)

made.

Point-for-Point Reply to Reviewers

Deep MALDI-MS Spatial 'Omics guided by Quantum Cascade Laser Mid-infrared Imaging Microscopy

Contents

Revision	2
1. Editorial Comments	2
2. Reviewer #1	3
1.2. Too many things at once	3
1.3. Expression and Grammar	5
1.4. Weak IR park, smoothing	5
1.5. Preprocessing and derivative spectroscopy	6
1.6. Incorrect wording for segmentation and preprocessing.....	7
1.7. Acronyms.....	8
1.8. Figure legends and captions	8
1.9. Final comment	8
1.10. Code availability	8
3. Reviewer #2	10
2.1. Comment	10
2.2. Different imaging modalities for guidance.....	10
2.3. Omitting information by guidance	12
2.4. Accuracy of lipid identification	14
2.5. Comment 1	15
2.6. Comment 2	16
2.7. Comment 3	17
2.8. Comment 4	18
2.9. Comment 5	18
2.10. Comment 6	19
2.11. Other minor comments.....	19

Revision

1. Editorial Comments

*As you will see from the reports copied below, the reviewers raise important concerns. We find that these concerns limit the strength of the study, and therefore we ask you to **address them with additional work**.*

We would like to thank the reviewers for their important remarks and constructive comments.

In particular, we thank reviewer1 for this statement:

“This work obviously is based on a tremendous amount of challenging work. I believe that after fixing the unfortunate factual and textual issues, it will be great contribution to an emerging field”.

To address criticism and suggestions by both reviewers, we have **performed a substantial number of additional experiments**. Moreover,

- to provide more focus on QCL-based MIR imaging-guided MALDI-timsTOF imaging, we have **removed the section on magnetic resonance MS (MRMS) imaging**. We have highlighted the fact that the on-tissue MS/MS technology that we used in TIMS-MSI is, in fact, **arguably the first MALDI imaging use of prm-PASEF (here: iprm-PASEF for imaging)**, the parallel fragmentation analysis of multiple precursor ions known from timsTOF proteomics;
- Instead of the MRMS data and to address “**new biology**”, we have added new experimental data on MIR imaging-guided single cell MS imaging of motor neurons in spinal cord and on **MIR imaging-guided analysis of multiple sclerosis-like lesions in mouse spinal cord**. We have added four co-authors in the course of this work. Perhaps most importantly, using our QCL-MIR imaging-guided TIMS-MSI with iprm-PASEF workflow, we **identified candidate proinflammatory markers of lipid remodeling** in this mouse model;
- in order to demonstrate the strength of the QCL-MIR imaging-guided TIMS-MSI with iprm-PASEF approach, we have comprehensively **characterized >150 sulfatide species by on-tissue MS/MS** (about 30 more sulfatides added than in the original manuscript with more than 100 additional on-tissue MS/MS spectra);
- based on additional experiments, we have added numerous explanatory, illustrative and supporting figures (one additional main Figure, seven additional supplementary figures and ten updated supplementary figures) and sub-figures;
- finally, we have substantially rewritten the manuscript to properly accommodate all changes made and all reviewers’ requests. Because the extensive editing might make the revised manuscript difficult to read, we also enclose a CLEAN version without edits.

2. Reviewer #1

The manuscript “Deep MALDI-MS Spatial ‘Omics guided by Quantum Cascade Laser Mid-infrared Imaging Microscopy” by Gruber et al., discusses the combination of infrared imaging and MALDI for spatial ‘OMICS. The combination is interesting. While there has been some work combining infrared imaging based on FTIR microscopy and MS, I don’t know many prior publications using EC-QCLs for imaging in combination with MS but now that Bruker (a co-funder of this work) also sells QCL microscopes, we can likely expect a more concerted push towards popularizing the combination. Unfortunately, the work has some significant issues and – in my opinion – should not be published without significant revisions.

We thank the reviewer for considering our combination of mid-infrared imaging and MALDI MSI for deep lipidomics profiling as “interesting” and for expecting “a more concerted push towards popularizing the combination [of EC-QCLs for imaging in combination with MS]”. We agree with this expectation. Our point-for-point comment is listed below.

1.2. Too many things at once

Above all, it appears, the authors main concern lies on demonstrating that they did everything there is to do in coupling mid-IR laser spectroscopy and MALDI. The manuscript wants to do too many things at once and thus cannot spend sufficient time on any single aspect of the research to be worthwhile to the reader.

We have addressed this important comment by focusing the storyline on the coupling of QCL-based MIR imaging microscopy and TIMS-MSI for the clear purpose of systematic on-tissue MS/MS fragmentation analysis for comprehensive compound annotation using the new imaging prm-PASEF (iprmPASEF) approach that is based on prm-PASEF widely used in the proteomics community. We describe a novel, coherent workflow that combines fast QCL-based MIR imaging microscopy for rapid acquisition of biological tissue sections and subsequent region-of-interest (ROI) identification with TIMS-MSI and iprm-PASEF. We believe that this workflow could become one of the standard approaches in spatial biology, as it combines the best of two worlds (MIR- and MS imaging).

It was and is not our strategy to do “everything there is to do” with this combination of technologies. Instead, we want to demonstrate the versatility and perhaps generalizability of the method by showing multiple examples without doing too much.

In order to focus, we have removed the figure and text relating to FT-ICR magnetic resonance MSI (**Old Fig. 2 and Old Suppl. Fig 13 and 14a**, and data from corresponding **Suppl. Tables**).

Our rationale for the revised storyline is as follows:

- We demonstrate that QCL-based MIR imaging microscopy is a **versatile tool for rapid data acquisition** from tissue sections **using spectral information that can be used for relevant ROI definition**. To highlight the power of this approach, we provide examples of segmentation of brain tissue (**Fig. 1a**), kidneys (**old Suppl. Fig. 12, now Suppl. Fig. 15**), investigated the morphology of 3D-cultures (**Fig. 1b**), and performed guided acquisition of functional tissue morphologies like kidney glomeruli (**old Fig 1ef, now Fig. 1e, now Suppl. Fig. 8 + 9**) down to single-cell segmentation of neurons (**new Fig. 1f; new Suppl. Figs. 10 and 11 based on new experiments**). All of these cases (and certainly others) can be analyzed in-depth by TIMS-MSI with iprm-PASEF.
- In order to demonstrate that our methodology is generalizable, we analyzed **two cases in depth with QCL-MIR-guided TIMS-MSI with iprm-PASEF analysis**: A) we identified many additional lipids in the ARSA-/- mouse model of human metachromatic leukodystrophy that had not been reported before (in total >150 sulfatides were identified directly in tissue) and used this extensive information for structure-CCS-relationship studies and B) (**new data**) we identify proinflammatory lipids specific for degenerative lesions in the experimental autoimmune encephalitis (EAE) mouse model of multiple sclerosis.

Please, note that we generally regard true compound identifications directly in tissue (based on MS/MS above all other things, but also based on *m/z* and CCS) as a prerequisite for any new biology discoveries. Therefore, we hold the view that a generic MIR-guided workflow that enables systematic on-tissue MS/MS is fundamentally important.

In going much deeper on these two topics, we trust that we now satisfy the reviewers very good request to “spend sufficient time on (this) research to be worthwhile to the reader”.

Therefore, we made changes to the following figures and tables:

We have added two more examples to highlight the versatility of the approach. To this end,

1. **Old Figs. 1f and 1g** were moved to the **Suppl. Fig. 9ef**.
2. **New Fig. 1f** and **Fig. 4** was created.
3. **New Suppl. Fig. 10 and 11** was created with corresponding **new Suppl. Tables 14-19**.
4. Method, Material and Supplemental Material sections were updated.

We have provided addition details on the iprm-PASEF analysis:

1. **Old Fig. 2b (now Fig. 2a)** and **2c** were generated with timsTOF instead of MRMS data and in particular **old Fig. 2c** was extended to provide an in-depth visualization of the imaging prm-PASEF method (**now Fig. 2b**) within our approach.
2. **Old Fig. 3b and 3c** were moved to the supplement (**now Suppl. Fig. 17c and 17d**) and replaced by a detailed analysis of the same example to demonstrate the benefit of long tims ramp times to yield a higher degree of non-chimeric MS² spectra for a given species (**new Fig. 2c and 2d**). Accordingly, the chemical structures of both molecules presented in **old Fig. 3d** were moved to

the supplement (**now part of Suppl. Fig. 17d**) to provide a more focused representation of the figure.

3. The supporting comparison between MALDI MSI and LC-MS data (**old Fig. 3a**) was updated and moved to the suppl. (**now Suppl. Fig. 22**) to focus on the imaging part in **Now Fig. 2**.
4. **Old Suppl. Fig. 1** was updated accordingly.
5. **Old Suppl. Fig. 19ij**, **old Suppl. Fig 13** and **old Suppl. Fig. 14a** were removed. Thus, **old Suppl. Fig. 14b** is now **Suppl. Fig. 16**.
6. **Old Table 1** and **Old Suppl. Tab. 13** were updated.
7. **New Suppl. Figs. 20 and 21** were added. Therefore **old Suppl. Fig. 19a** was removed. **Old Suppl. Fig. 19b** is now **19**.

Related to the CCS-value analysis of sulfatides identified in the ARSA-/- mouse model, we made the following changes:

1. **Old Suppl. Tab. 15** is **now Table 2**, which was updated accordingly to highlight the results of the *structure-CCS-relationship* investigations.
2. **Old Fig. 4** is **now Fig. 3** with the following modifications: **MALDI data was updated**.
3. **Old Fig. 4c (now Fig. 3c)** now contains also data from the new sulfatides found and their relative deviation. Accordingly, **old Suppl. Fig. 20a and b (now Suppl. Fig. 24a and b)** were updated.
4. Due to the large amount of sulfatide species, a **new Suppl. Fig. 28** containing information on the structure-CCS-relationship was added.
5. **Old. Suppl. Fig. 25** is **now Suppl. Fig. 31** and contains now **a** and **b** part including chemical structures.

1.3. Expression and Grammar

English expression and grammar is weak and many nonsensical sentences remain. Figures appear to be present just to fill space or provide eye candy without conveying meaningful information. In some places the manuscript is unintentionally(?) funny. For example, the phrase "at very mass resolution" (pg. 10, ln. 274) would be more at home on a picture of a Shiba Inu than in a scientific manuscript.

We reviewed English expression and grammar and updated figures throughout the manuscript (see comment 1.2).

We ensured that all figures presented in the manuscript and supporting information are of major importance to support our scientific results and the innovation presented in the paper.

1.4. Weak IR park, smoothing ...

Overall, the mid-infrared spectroscopy part of this manuscript is weak. For example, pg 22 ln 606 claims that "asymmetric least square[sic] smoothing [was used to] correct baseline distortion and light-scattering effects" Leaving aside that it might be more fitting to cite Eiler's original work rather

then one of the authors' previous papers, asymmetric least-squares based baseline correction cannot correct light-scattering effects. A method like Bassan's RMieS-EMSC would be required to do this, asymmetric least-square only corrects the baseline offset, one of the effects of scattering.

We have upgraded the IR part substantially by adding citations and by revising the main manuscript to provide more specific information.

1. The corresponding part of the manuscript was re-written and citations were added (line 658): “Subsequently, data pre-processing such as baseline correction and spectral differentiation was performed based on case-specific, user-defined settings. Typically, baseline distortions of absorbance spectra were corrected using asymmetric least square smoothing [Eilers, P. H., & Boelens, H. F. (2005) , PMID: 29321555] (smoothing factor 1,000,000, weighting factor 0.01 and 10 iterations as default) and piecewise linear interpolation was utilized to calculate the first and second order derivative. Other pre-processing algorithms commonly used in MIR spectroscopy like resonant Mie scattering correction [PMID: 30793501] could be integrated in the future. “
2. We have added more information and citations to highlight how QCL technology has led to a paradigm shift in MIR imaging, paving the way for new applications (line 94): “In contrast, quantum cascade laser (QCL)-based MIR imaging microscopes feature a tunable coherent light source with high power-density for high sensitivity and higher sample throughput in biological systems¹⁴⁻¹⁸, thus enabling multiple new technologies and applications in MIR imaging utilizing vibrational probes(PMID: 32601425 , PMID: 38191490), large scale plasmonic metasurfaces (PMID: 25836797) or MIR-based whole slide scanning (PMID: 37626026).”
3. We have added a **new Suppl. Fig. 5** to show data on MIR lipid fingerprints, in particular of a sulfatide brain mixture, to provide further information on a sulfatide MIR fingerprint and thus to underpin the chosen set of wavenumbers used during the segmentation process.
4. All figures and images were revised and updated to contain labels and data points whenever they were missing before.
5. We have added a new application (**new Fig. 4 and new Suppl. Figs. 10 and 11**) and updated the “Feature-selective image segmentation for definition of regions-of-interest (ROI)” part (line 671ff) of the manuscript containing information of use case 4 (neurons in spinal cord) (line 697ff) and use case 6 (lesions in white matter of spinal cord) (line 716).

1.5. Preprocessing and derivative spectroscopy

In general, spectral preprocessing is not explained at the depth that would be required. For example, the authors calculate second derivatives of spectra but do not explain how this derivative is carried out. Was Savitzky-Golay used? Which parameters?

Two methods, a linear interpolation between adjacent data points and Savitzky Golay's method, were implemented for spectral differentiation. The methods section was updated to be more specific, e.g. in line 661 we added

"... and piecewise linear interpolation was utilized to calculate the first and second order derivative. For higher order polynomial interpolation and spectral differentiation, a Savitzky-Golay filter was implemented using the sciPy package [PMID: 32015543]".

Furthermore, we added the information on the used parameters (line 717)

"Second derivatives of spectra were calculated using a Savitzky-Golay filter with a third-degree polynomial and a window length of 7."

1.6. Incorrect wording for segmentation and preprocessing

Sentences like "QCL-IRI spectral features discriminating between tissues, tissue morphologies, or cell types are then extracted from the IR "fingerprint" spectra and used to segment QCL-IRI data, e.g. by performing 2nd derivative spectroscopy (Fig. 1a(ii)), and to define ROIs (Fig. 1511a(iii))." appear throughout the work. Sentences, that have obvious grammatical issues but are also factually incorrect: calculating the second derivative is not an algorithm used for segmentation but for preprocessing mid-IR spectra.

Thanks for pointing this out. We changed some sentences, like the one mentioned above, throughout the manuscript.

E.g.

"QCL-IRI spectral features discriminating between tissues, tissue morphologies, or cell types are then extracted from the IR "fingerprint" spectra²¹ and used to segment QCL-IRI data, e.g. by performing 2nd derivative spectroscopy²¹ (**Fig. 1a(ii)**), and to define ROIs (**Fig. 1a(iii)**)."

is now (line 165)

"After data pre-processing, distinct spectral features were selected from the MIR fingerprint region and corresponding images were used for image segmentation and definition of morphological regions-of-interest by, e.g., k-means clustering (PMID: 32555465) (**Fig. 1a(ii)** and **Fig. 1a(iii)**)."

Or e.g.

"In step 2, these sets of wavenumbers were employed for k-means clustering to define ROIs."

is now (line 676)

"In step2, k-means clustering for ROIs definition was utilized on a reduced hyperspectral data cube defined by a selected set of spectral features, if not stated otherwise."

1.7. Acronyms

The work introduces acronyms like "QCL-IRI" that are not used in the mid-IR community but only in this manuscript.

We thank the reviewer for this remark.

We decided to use the following wordings in agreement with literature:

"QCL-based MIR imaging microscopy-guided MALDI MSI" (short: "QCL-MIR imaging-guided MSI"). Consistently, figures now contain "MIR" labels, whenever MIR imaging microscopy data is presented or was used for data evaluation.

1.8. Figure legends and captions

The figures depict infrared data in way that makes it essentially useless (no units on axes, interpolation without showing actually measured points). In Fig. 1 a (iii), a(vi), f,g, i, Fig. 2b, Fig. 3C images have no color bar and the depicted information is thus meaningless. The same is repeated for figures in the supplementary. The color scheme in Fig. 2c is puzzling. According to figure caption and legend three groups (I: orange, II: red, III: green) are depicted. However, the image itself does not contain solid colors but gradients. It is not clear that the gradient colors mean or, for some of the hues, which of the three groups they should be assigned to.

We have revised all figures:

1. Measured data points in **now Suppl. Fig. 5a(ii), 5b(ii), 8c(ii) and 13b** were added.
2. We added color schemes, clustering labels and scale bars wherever missing.

As stated in the main manuscript (line 668), "For simplicity, we consider the data obtained from spectral differentiation, usually given in units of $1/\text{cm}^{-1}$ (1st derivative) and or $1/\text{cm}^{-2}$ (2nd derivative), as unit-less", since it does not provide any physical meaning. Intentionally, (semi)-schematic figures of absorbance data in **now Fig. 1(i) and (iv)** do not contain units, ticks and tick labels similar to the figure in **now Fig. 1(iv)** for the MS^2 spectra.

1.9. Final comment

In closing, this work obviously is based on a tremendous amount of challenging work. I believe that after fixing the unfortunate factual and textual issues, it will be great contribution to an emerging field.

We thank the reviewer for this very positive feedback on our work.

We believe that we have provided a comprehensive revision, such that the manuscript can now be considered for publication in Nature Communications.

1.10. Code availability

The authors promise to share code in the abstract, but I was not able to find any code with the provided materials.

Gruber, Schmidt et al. **Reply to Reviewers**

Code and executable-file of our software tool was shared with the editors and thus available during the review process.

3. Reviewer #2

2.1. Comment

In this manuscript, Gruber et al. employed QCL-IRI-guided MSI technology to confine highly interested areas for spatial lipidomics profiling at maximized analytical depth with MR-MS or PRM-PASEF-MS analysis. The authors demonstrated the applications of the technology in various types of samples such as cells and tissues, illustrating the advantages of the technique in deep lipidomics profiling.

Thank you.

2.2. Different imaging modalities for guidance

However, the uses of different imaging modalities for selective MSI sampling have been already described by several publications, making the manuscript less innovative (e.g., <https://doi.org/10.1021/jasms.0c00128>, <https://doi.org/10.1021/acs.analchem.5b03610>, <https://doi.org/10.1021/acs.analchem.8b02885>, <https://doi.org/10.1039/C5SC03782D>). Moreover, the IRI-guided MSI strategies have been also previously published according to the authors (<https://doi.org/10.1038/s41598-017-18477-6>), only the tool used to find ROIs was changed.

We agree with the reviewer that this is not the first manuscript introducing an imaging guiding modality for subsequent MSI analysis. In fact, as pointed out by the reviewer, we have introduced such an approach ourselves, related to FT-IR-based MIR imaging guided MSI (Rabe *et al.*, 2018, cited in this manuscript). Most studies referred to in reviewer #2's comment, were cited in the Rabe *et al.* paper.

However, introducing a hyperspectral modality for guided MSI is not a major aspect of innovation in THIS manuscript. We don't claim that this is the first paper presenting a guided MSI workflow. We'd be happy to cite more of these studies in this manuscript, but are already exceeding the limit for number of citations.

In this manuscript, we **use rapid QCL-based mid-infrared laser imaging microscopy for MSI guidance for the first time**. This can be up to 175x faster than the FT-IR-based approach employed by Rabe *et al.* This could provide the basis for automation and, most importantly, lets users re-invest the time saved by MIR-based segmentation for default ion mobility-based on-tissue MS/MS using iprm-PASEF. In that sense, we believe that the **novelty of our study is the entire workflow QCL-MIR imaging-guided TIMS-MSI (in tims-on mode) and iprm-PASEF for on-tissue MS/MS**.

Heijs *et al.*, referenced above by the reviewer, for example used manual drawing of ROI prior to MSI, Garrard *et al.* manual drawing with a computer mouse; Patterson *et al.* used automated interpretation of autofluorescence images but made no statement about speed. As addressed in the manuscript, previously published work using a distinct guidance modality for on-tissue analysis including our own "remained a mere concept so far" due to technical limitations.

Besides much higher speed, the richness of mid-infrared spectrum offers more versatility than, for instance, polarimetry that the reviewer refers to and that we cited in Rabe *et al.* as a pioneering study.

It also does not require per se outside input like histopathology annotations. In that sense, QCL-based mid-infrared laser imaging microscopy is a special guidance modality.

We exemplify the enormous potential of our workflow for in-depth analytical chemistry with default MS²-based compound annotation directly on tissue using the new iprm-PASEF method. No other paper that we know of combines any guidance modality with MSI and on-tissue (!) MS²-based identification of >150 compounds. Using this approach, we were able to investigate and analyze >10 brain sections, >30 spinal cord tissue sections, >150 spheroids and >100 kidney sections in one study.

In summary, our innovation claim is NOT to be the first to ever use guidance for MSI, but to provide a unique workflow (QCL-MIR guidance → ROI-focused MSI → in-depth on-tissue lipidomics with iprm-PASEF → systematic studies of structure-CCS relationships) and computational tools as a community resource.

To this end, we have focused the manuscript on the QCL-based MIR imaging microscopy-guided TIMS-MSI approach including iprm-PASEF and removed the MRMS data from the manuscript. This includes the following modifications:

1. **Old Fig. 2b (now Fig. 2a) and 2c** were generated with timsTOF instead of MRMS data and in particular **old Fig. 2c** was extended to provide an in-depth visualization of the imaging iprm-PASEF method (**now Fig. 2b**) within our approach.
2. **Old Fig. 3b and 3c** were moved to the supplement (**now Suppl. Fig. 17c and 17d**) and replaced by a detailed analysis of the same example to demonstrate the benefit of extended tims ramp times to yield a higher degree of non-chimeric MS² spectra for a given species (**new Fig. 2c and 2d**). Accordingly, the chemical structures of both molecules presented in **old Fig. 3d** were moved to the supplement (**now part of Suppl. Fig. 17d**) to provide a more focused representation of the figure.
3. The supporting comparison between MALDI MSI and LC-MS analysis (**old Fig. 3a**) was updated and moved to the suppl. (**now Suppl. Fig. 22**) to focus on the imaging part in **Now Fig. 2**.
4. **Old Suppl. Fig. 1** was updated accordingly.
5. **Old Suppl. Fig. 19ij, old Suppl. Fig 13 and old Suppl. Fig. 14a** were removed. Thus, **old Suppl. Fig. 14b** is now **Suppl. Fig. 16**.
6. **Old Table 1 and old Suppl. Tab. 13** were updated.
7. **New Suppl. Figs. 20 and 21** were added. Therefore **old Suppl. Fig. 19a** was removed. **Old Suppl. Fig. 19b** is now **19**.

This also further strengthen our results related to structure-CCS-relationships, for which we made the following changes to highlight the strong statistical and analytical basis of the results:

1. **Old Suppl. Tab. 15** is now **Tab. 2**, which was updated accordingly to highlight the results of the *structure-CCS-relationship* investigations.
2. **Old Fig. 4** is now **Fig. 3** with the following modifications: **MALDI MSI data was updated**.
3. **Old Fig. 4c (now Fig. 3c)** now contains data from the new sulfatides found and their relative deviation. Accordingly, **old Suppl. Fig. 20a and b (now Suppl. Fig. 24a and b)** were updated.

4. Due to the large amount of sulfatide species, a **new Suppl. Fig. 28** containing information on the structure-CCS-relationship was added.
5. **Old. Suppl. Fig. 25** is now **Suppl. Fig. 31** and contains now **a** and **b** part including chemical structures.

By combining i) QCL-based MIR microscopy with ii) TIMS-MSI and iii) imaging prm-PASEF analysis, we believe we have established a novel high-content concept for in-depth bioanalysis of biological specimen in a single workflow that uses new hardware and software and that we make available to the scientific community by providing a home-build computational tool.

2.3. Omitting information by guidance

Most importantly, MSI technology is developed for comprehensive spatial-omics profiling. MSI technology is featured to explore molecular features that other technologies such as immunohistochemistry and IRI cannot differentiate. Thus, relying on ROIs defined by IRI for MSI-based spatial profiling would not support novel biological discoveries.

Having worked in the MSI field for 15 years, we certainly agree with the reviewer that “MSI technology is featured to explore molecular features that other technologies such as immunohistochemistry and IRI cannot differentiate”. That is the unique selling proposition (USP) of MSI.

“Molecular feature” and resulting novel biological discoveries in our opinion refers (at least) to three aspects:

1. Certainty of the molecular identification: **A molecular feature that is merely claimed, but that is not truly supported by analytical chemical data, at best provides a basis for a biological hypothesis.** And it is the “molecular identity” meaning of “molecular feature” that sets MSI apart from other technologies. MIR imaging may address a similar number of “molecular features” (here: wavenumbers corresponding to vibrations of molecular bonds) as MSI does, but it will never be able to aid molecular identification.
However, MSI is currently lacking in “molecular identification”, because most studies still use low resolution MALDI-TOF MSI and several studies use FTMS-MSI (not always FDR-controlled) with much better resolving power and mass accuracy than MALDI-TOF but with very little if any on-tissue MS/MS capabilities. Currently, even TIMS-MSI studies rarely use the TIMS function, because it slows down data acquisition considerably (up to 10-fold). We’d like to argue that **the workflow presented in our manuscript can open the door for routine use of TIMS and on-tissue MS/MS as a technological basis for truly (analytically speaking) novel biological discoveries** rather than mere claims thereof.
2. Statistics for that molecular identification: If a molecular feature is observed in an N=1 experiment, it does not yet imply a biological discovery.

Also here, MSI is currently lacking, as many published studies are currently N=1 because of time constraints. We'd like to argue that the workflow presented in this manuscript can open the door for routine use of N>1.

3. Accurate or at least best possible definition of ROIs: The relationship between “molecular features” and ROI definition is not an easy one: First, also in MSI it could be a limited number of all molecular features (e.g. by peak picking followed by m/z features selection) that contribute to segmentation/ROI definition, e.g. based on k-means clustering. It is difficult to find reported numbers, but e.g. 228 lipids have been mentioned (<https://www.frontiersin.org/journals/chemistry/articles/10.3389/fchem.2023.1332816/full>). For principal component analysis (PCA), we have shown that only a small number of principal components contains relevant information. Loadings plots in PCA reveal limited numbers of m/z with meaningful contributions to PCA (doi: 10.1007/s00216-014-8356-9.). Second, in order to be useful for clustering/segmentation/ROI definition, repeatability of molecular features is more important than whether or not the features have molecular identities attached to them. This is true for many MSI studies that use non-annotated m/z values (i.e. numbers without names) for segmentation. For instance, many ROI definitions in MSI use MALDI-TOF data, which is typically devoid of molecular IDs.

In our opinion future side-by-side comparisons will have to investigate how MIR- and MSI-defined ROIs compare. In this manuscript we provide several examples (segmentation of brain tissue (**Fig. 1a**), kidneys (**old Suppl. Fig. 12, now Suppl. Fig. 15**), investigation of the morphology of 3D-cultures (**Fig. 1b**), performing segmentation and guided acquisition of functional tissue morphologies like kidney glomeruli (**old Fig. 1ef, now Fig. 1eold Suppl. Figs. 8 and 9**) down to single-cell segmentation of neurons (**new Fig. 1f; new Figs. 10 and 11** based on new experiments) that suggest that they compare rather well.

This comparison is complicated by the fact that ground truths are lacking for spatial segmentation methods, as pointed out by Theodore Alexandrov (<https://www.annualreviews.org/content/journals/10.1146/annurev-biodatasci-011420-031537>): “For example, numerous spatial segmentation methods have been developed (<https://analyticalsciencejournals.onlinelibrary.wiley.com/doi/full/10.1002/mas.21602>), but without ground truth data, comparing them is highly subjective, because it is not known which spatial regions in a tissue section have distinct molecular compositions.”

In **old and now Figs. 1be, new 1f; Suppl. Figs. 8ab, 9a-d, 13a and 15; new Suppl. Figs. 10 and 11** we present data to suggest that image segmentation performed based on the MIR data mimics the tissue morphology. A potential partial loss of information (not all glomeruli detected, or deviation of the ISOM segmentation compared to MSI segmentation is on a few % level) is compensated by the information throughput for subsequent statistical analysis.

In a nutshell: Molecular features as defined by MSI are not necessarily better or more useful for segmentation/ROI definition than the ones defined by MIR imaging. It is the unique ability of MS to eventually (but this can be done after ROI definition) identify the exact chemical nature of biomolecules that sets it apart. Since the workflow we present here does exactly that, we believe that

it will strongly promote discovery of novel biology by providing more molecular identifications and larger N.

In our manuscript, we demonstrate that fast ROI definition by MIR imaging in combination with ion mobility MSI (timsTOF) AND the iprm-PASEF approach for the first time enables MS²-based identification of >150 lipids directly on tissue. Hence, biomolecular discoveries are done by MSI in improved fashion, but guided by MIR.

As pointed out in comment 2.2, we removed **old Fig. 2** (MRMS imaging), in order to focus solely on timsTOF-MSI with iprm-PASEF. Instead, we added a new biological example (**new Fig. 4, new Suppl. Fig. 10 and 11, and new Suppl. Tables 14-19**) on lipid profiling in the EAE model of multiple sclerosis, in order to demonstrate HOW this approach does supports the discovery of new candidate spatial biomarkers, e.g. the proinflammatory ceramide-1-phosphate. Furthermore, we demonstrate that mid-infrared imaging microscopy can successfully segment out individual neurons, the content of which is then explored by MSI, of course. This might be important for future motor neuron disease studies.

Our revised lipid analysis in the ARSA mouse model, showed, that about 5 sulfatide species (3%) could not be identified based on MSI analysis on the ISOM and IMP region, since they primarily accumulate in the kidney cortex. This information would be lost as a consequence of focusing on ISOM/IMP, but it would also be lost if these ROIs were defined by MSI. However, as demonstrated in **Table 1**, we found 45 additional sulfatides in our guided setting compared to LCMS of whole kidney extract. Based on that, the loss of these five cortex sulfatides seems to be rather reasonable. A potential partial loss of information is also compensated by increased information throughput.

2.4. Accuracy of lipid identification

Additionally, the accuracy of lipid identification needs further improvement in this work. More statistical data should be provided to demonstrate the advantages of the strategy used in this work.

We agree with the reviewer that providing accurate lipid identifications is indispensable in bioanalytical science and beyond. Using our approach, we provide accurate m/z values as well as CCS values and their uncertainties for all lipid species identified using many technical and/or biological replicates (**Suppl. Table 1, Suppl. Table 10 and updated Suppl. Table 13**). Measured m/z values for the timsTOF MALDI MSI are governed by mass deviations below 3 ppm. Furthermore, during revision we have **performed additional iprm-PASEF experiments, and now provide MS² data for a large number of lipids (>150), in particular sulfatide species** using the ARSA-/- mouse model. To our knowledge, this is the most comprehensive MS²-based study on an individual lipid class in MSI so far. By structure elucidation of the detailed fragmentation patterns, we obtained analytical sound lipid identifications. To strengthen the quality of our data, we updated **old Suppl. Fig. 19 and added new Figs. 20 and 21**, where all identified sulfatides are classified by their fragment patterns and further classified as non-chimeric, i.e. clean, or chimeric. In total, the fragmentation analysis was performed on about >200 lipids (considering all studies). Fig. 3a, c and **d** were updated (see also comment 2.2 for further

modifications performed in the revision process). In summary, within this manuscript we provide an unmatched number of identifications on-tissue that is based on highest quality at different analytical levels (m/z , CCS, MS^2). To accomplish this, the high information throughput was indispensable, as more than 100 kidney samples and more than 150 spheroids samples were measured and considered during the data evaluation, as demonstrated in, e.g., **Fig. 1d**. Besides large sample numbers, statistical soundness is supported by various statistical tests, e.g., t-test used for the Volcano plots or Wilcoxon Rank-Sum Test.

We have also added dedicated figures (**Fig. 2c and 2d**) and provided more information in the method section to demonstrate the benefit of using long ramp times for the tims technology to yield a higher degree of non-chimeric MS^2 spectra for given species. The use of long ramp times is not uniquely to this workflow, however it became practically using this approach and further improves the data quality and therefore the fragmentation pattern analysis.

2.5. Comment 1

The procedure for lipid identification was unclear. If I understand correctly, for data acquired by TIMS-MSI based prm-PASEF fragmentation, only exact mass match was applied for annotation of sulfatide in ARSA -/- mouse kidney (Table 1). To achieve deep lipidomics profiling, it is better to use multi-dimensional information such as CCS and MS/MS match for high-confident lipid identification. For data acquired by MR-MSI, exact mass match and IFS analysis improved identification confidence. However, only 34 and 39 out of 91 and 97 spectra were supported by IFS information. How does the author interpret the annotations that do not conform to IFS analysis? I want to know whether the isobaric overlappings among different lipid species interfered IFS analysis, such as $[M+2]$ isotopologue of SM4 32:2;O2 and M0 of SM4 32:1;O2. For data acquired by LC-TIMS-MS, 4D-descriptor match was applied for lipid annotation as described in Methods. It is better to provide more details for the establishment of 4D-descriptors.

We thank the reviewer for the detailed questions regarding the IFS analysis. The review process prompted us to focus on the TIMS-MSI data analysis only, and **we therefore removed the MRMS-MSI section entirely**.

As demonstrated in **Suppl. Table 1-19**, to achieve deep lipidomics profiling we have considered multi-dimensional information (m/z , CCS, MS^2 and for LC-MS also RT) throughout the manuscript. We clarified this in the manuscript in the "QCL-QCL-MIR imaging-guided trapped TIMS-MSI enables extensive fragmentation analysis on tissue and sulfatide identifications on par with LC-based 4D-TIMS-PASEF" section and added sentences in e.g. line 78.

For LC-TIMS-MS analysis, we updated the methods section with a detailed description of the establishment of the 4D-descriptors. Briefly, the data acquired from ARSA-/- and ARSA+/+ samples were uploaded to Metaboscape 2021b, which extracted all features that matched the processing method parameters. Each feature had associated m/z , RT, CCS, and/or MS^2 spectra. Following manual

annotation of sulfatides based on their MS² spectra, the identified sulfatide species along with their 4D-descriptors (m/z , R_T , CCS, and MS² spectra) were used to establish the **Sulfatide Analyte List**.

2.6. Comment 2

Page 6 line 169 & Page 8 line 221: The authors used sets of wavenumbers to define ROIs and validated the feasibility of the QCL-IRI with other techniques such as multiplex-MALDI-IHC and MSI. I would like to know if there are guidelines for choosing case-specific sets of wavenumbers in QCL-IRI analysis for various types of samples. How to ensure that the chosen areas are indeed of interest? If additional methods are required for verification each time, the applicability of the technology may be compromised. Additionally, these applications relied on prior knowledge, suggesting that the technology may have limitation for new biological discovery.

In our opinion, QCL-MIR imaging is not much different than MSI in this respect. Also, in MSI ROIs are frequently either defined based on prior knowledge (e.g. presence of a drug or a known lipid/metabolite marker) or in a two-step process: First a set of possible markers or an m/z signature is discovered in small sample sets or the best segmentation algorithm (e.g., selection of k in k -means clustering) is selected – all by comparison to some kind of ground truth like histopathology annotations, autofluorescence images etc. Then – after validation against the ground truth that demonstrates relevance, this marker set, signature or segmentation algorithm is applied to larger sample cohorts. In that sense, “verification each time” means that for every new use case additional methods are required also in MSI. We are not aware of a generic way to define, for instance, viable tumor lesions, by MSI without additional reference methods. In our hands, MSI signatures for glioma have not translated to even other types of tumors. We have frequently segmented very heterogeneous tumors with many computational methods, but we know no way of telling, which segment is tumor, without additional reference methods. Even machine learning (ML) methods are typically supervised, i.e. they require labels that can be used during training (e.g. Enzlein et al. Anal Chem. 2024 Jun 18;96(24):9799-9807): “The integration of MSI data with ML-based feature extraction further revealed that plaque-associated gangliosides GM2 and GM1, as well as A β 1-38, but not A β 1-42, are relevant differentiators between the investigated pathologies.”). Along the same lines, Esselman et al suggest (<https://www.biorxiv.org/content/10.1101/2024.02.21.581450v1>): “Additional imaging modalities can be integrated with MALDI IMS to associate these biomolecular distributions to specific cell types. Herein, we demonstrate an integrated workflow combining MALDI IMS and multiplexed immunofluorescence (MxIF) microscopy.”

Multimodal approaches are frequently used for cross-validation and/or correlative data analysis. To provide solid ROIs and a high level of confidence, we have acted accordingly (**Fig. 1b, old Figs. 1 ij, now Figs. 1gi, new Fig. 1h, Suppl. Figs. 8 and 9, new Suppl. Fig. 10, now Suppl. Figs. 13a and 15**). However, it is worth mentioning, that this cross-validation might be done only once during the phase of designing an experimental campaign to target a biological question, similar to method development in MSI (see above).

We have built an additional safe-guarding feature into our workflow: ROIs defined by MIR imaging can be computationally dilated prior to MSI (**Suppl. Fig. 9c**). That way MSI-ROIs are larger than MIR-ROIs, and the MSI data can be used to examine if the MIR-defined ROI boundary is visible in MSI, i.e. if there is a molecular correspondence in MSI.

Defining ROIs with IRI analysis, which provides limited information, for further MSI analysis may result in the loss of valuable information.

There are several aspects to an answer, and we'd like to refer the reviewer and editor to comment 2.3.

The authors should provide new demonstration examples to support new biological discovery instead of validation with known examples.

We took this point very seriously and demonstrated how our method, in principle, supports new biological discoveries using a murine animal model of multiple sclerosis (MS), experimental autoimmune encephalitis (EAE) (**new Fig. 4, new Suppl. Figs. 10 and 11, and new Suppl. Tables 14-19**) in collaboration with the Medical University in Vienna. We have added four new co-authors in the process.

Our results revealed the following new biological insight (line 483ff):

MIR imaging microscopy-guided TIMS-MSI of EAE spinal cord lesions (versus white matter in control spinal cords) revealed new molecular features of lipid remodeling. Most importantly, our comprehensive TIMS-MS/MS analysis identified CerP 34:1;O2 and CerPE 36:1;O2 as key lipids associated with EAE pathology. CerP had been implicated in multiple sclerosis before, but its localized strong upregulation (>20x) was unknown. Finding relevant molecules in a spatially resolved manner validates our approach. Little is known about CerPE, but its localized >20x upregulation in lesions (just like CerP) suggests that it may be another biomarker like CerP. Validation of this finding will require extensive follow-up studies.

Besides demonstrating applicability for new biological discoveries, we made careful validation of this complex workflow with the known ARSA^{-/-} example another priority (see above). New biological discoveries require true molecular identifications based on high quality, multidimensional data. The method proposed in the manuscript is in general designed to provide both multidimensional data and molecular identifications.

2.7. Comment 3

Page 10 line 271: In this part, the authors combined QCL-IRI with MR-MSI for confident lipid identification. In Figure 2, the authors emphasized the advantages of ultra-high resolution MR-MSI. But the assistance provided by QCL-IRI was not adequately explained in the main figure. It is not novel to use ultra-high resolution mass spectrometry and FDR-controlled analysis for lipid identification (<https://doi.org/10.1126/science.abh1623>).

As pointed out above, we have decided to focus the manuscript on combining MIR with TIMS MSI to provide a more focused study.

2.8. Comment 4

Page 14 line 373: The authors focused to the ISOM and IMP ROIS using TIMS-MSI with 2D-mobilogram-based prm-PASEF, which provided additional dimensions for lipid identification. More detailed description of the prm-PASEF method is needed (e.g., how many MS2 spectra were collected for each pixel; how was the prm-list generated). In addition, were product ions used for quantification in prm-PASEF method? Additionally, evaluations of the quantitative accuracy and reproducibility are needed.

We thank the reviewer for this comment. We included a more detailed description of the iprm-PASEF setup in the manuscript and the methods part and included a detailed workflow description and data in **new Fig. 2b (old Fig. 2c)**. Line 334ff: For each iprm-PASEF measurement, consisting of a maximum of 15 precursors, 883 pixels were investigated in IMP (8 min) and 1554 pixels in ISOM (13 min) at 40 μm lateral step size. The precursor lists were generated in order to avoid overlaps in the mobility windows of the respective precursors. Further **new Suppl. Fig. 20 and 21 (old Suppl. Table 14)** were introduced, and the results are further included in **new Fig. 2g and new Suppl. 22 (old Fig. 2a and old Suppl. Fig. 15)**. No quantification with product ions was carried out, since this was not a qMSI study. The reproducibility is given by our N=4 approach, which means that all precursor ions were analyzed for each individual specimen for the ARSA-/- mouse model. We found our data to be highly reproducible (**updated Suppl. Tab. 13**). Upon acceptance of our manuscript, we will make all the data publicly available.

2.9. Comment 5

Page 14 Figure 3: The authors illustrated the advantages of TIMS separation in QCL-IRI-guided MSI via several examples. Several works have already demonstrated the advancement of TIMS in MSI field (<https://doi.org/10.1038/s41467-023-40512-6>, <https://doi.org/10.1038/s41467-023-43298-9>). It is necessary to highlight the combination of QCL-IRI with MSI analysis rather than simply illustrating the advantages of TIMS separation through examples. Additionally, supplementing statistical data rather than examples would enhance the reliability of conclusions.

To provide new data, we have added a **new Fig. 2b** to demonstrate the combination of MIR-guided MSI analysis using iprm-PASEF. For full coverage of the generated sulfatide database, iprm-PASEF was performed on the ISOM using 9 individual iprm-PASEF runs on 3-4 technical replicates to cover sulfatides predominantly present in the ISOM region. In the same way, sulfatide identification was performed on the IMP part (5 runs). This entire process was repeated on 4 biological replicates for statistical coverage. **New Fig. 2cd** shows an evaluation of the benefit of our long ramp times. Since the ability to use this long ramp times is a direct result of the time saving by guidance, it highlights the advantage of combining QCL-MIR imaging and MSI.

We now describe this in more detail in the manuscript (line 328ff).

New Suppl. Figs. 20 and 21 demonstrate an evaluation of all MS² spectra for the identified sulfatides. In addition, our data on the spheroid use case demonstrates the statistical power of the presented approach for high information throughput studies. For instance, in **Fig. 1d** data from the mean MSI profiles from >150 technical replicate spheroid sections were used, rather than presenting data for individual pixels. For the spheroid use case, the precursor selection was performed via LASSO (Least Absolute Shrinkage and Selection Operator) regression and their respective box plots are shown (**Suppl. Fig. 6**). The performance estimation for the glomeruli use case was performed on N=4 tissue sections (**Suppl. Fig. 8**). In line with all the other use case studies, the *m/z* feature finding is based on statistical processes included in volcano plots (**now Fig. 1i, new Fig. 4c; Suppl. Fig. 18a**) and box plots (**now Fig. 1i, now Fig. 4e; Suppl. Fig. 6**).

As discussed above, QCL-MIR guidance provides speed and enables higher throughput MSI focused on ROI or extensive on-tissue MS² using iprm-PASEF by re-utilizing the time saved.

2.10. Comment 6

Page 16 line 447: Structure-CCS relationships of experimentally annotated sulfatide lipids were explored. Further, predicted CCS values were summarized via experimental lipidomics data. As a resource, the information accuracy is very important. As mentioned in the 1st comment, it is necessary to provide accurate lipid identifications at first.

We thank the reviewer for pointing out that our sulfatide database and all other lipid identifications are of high interest for the community.

In addition to **now Suppl. Figs. 18-21**, we have investigated further (see comment 2.2 and 2.4) to highlight our accurate lipid identifications (**new Suppl. Figs 22 and 23**). Experimental uncertainties obtained for the CCS values are provided in **now Suppl. Tab. 13** and were updated. In addition, we have added an additional suppl. figure (new Suppl. Fig. 25) to provide a clearer comparison between the MSI results and the results from the prediction tools. **Experimental** uncertainties are easier visible to the reader and support our findings with respect to the prediction provided by AllCCS2. Further, residuals were added to **new Suppl. Fig. 28** and now Suppl. Fig. 29 and the results for the structure-CCS-relationship analysis were updated (**new Tab. 2; Fig. 3ac; now Suppl. Figs 24 and new 28**).

2.11. Other minor comments

1. *Page 12 line 317: Please provide details of cross-modality (semi-)quantitative analysis in Methods.*

We have updated the method section and **now Suppl. Fig. 13b-d**. We added a new **Suppl. Fig. 14** as well as highlighted the investigation of lipid accumulation by replacing former **Fig. 1j** with a new **Fig. 1h**. Former **Fig. 1j** appears in the Supplement as **Suppl. Fig. 5b**.

2. *The figures lacked essential labels, such as those in Figure 1c, Figure 1f, Figure 2i, and so on.*

We have revised all figures and figure captions.

3. *It is advised to carefully review the manuscript as there are several typos present. There is a lack of data in Supplementary Figure 22.*

We have revised the entire manuscript. There is no lack of data in **old. Suppl. Fig. 22. Old. Suppl. Fig. 22** (now **Suppl. Fig. 26**) is used to support the finding of **now Suppl. Fig. 3d and 3e** regarding the SM4 18+n:1;O3 , SM4 18+n:1;O2 , SM3 18+n:1;O3 , SM3 18+n:1;O2. Highlighted areas were added. As demonstrated in **now Suppl. Fig. 17**, SM2a species were not observed in LC-MS. Their values are described in **Suppl. Table 13**.

4. Please check the citation of the references. There might be an error in the citation of reference 48.

We have added references to the manuscript. Therefore, we have also addressed this remark.

We addressed the reviewer comments by updating Methods, Figures, the language and expression in general, and References.

Point-for-Point Reply to Reviewers 2nd Revision

Deep MALDI-MS Spatial 'Omics guided by Quantum Cascade Laser Mid-infrared Imaging Microscopy

1. Reviewer #1

2.1 Comment 2

I am happy to see that the authors have further improved the manuscript. While there may still be some points that seasoned IR experts might quibble about, this is one of the first papers in an emerging field and no doubt follow up work will be required to explore details.

We thank Reviewer #1 for valuing the additional work we put into the revised manuscript.

2.2 Comment 2

For future work, I can not emphasize enough the importance of considering standing wave effects and scattering effects in infrared spectral imaging. The field has painfully learned the strong effects scattering has on spectra and that it not only affects the baseline but also relative intensities in the late 2000s. Open software packages for correction exist. e.g. in the OpenVibSpec code. I also want to note that all of these effects are seen especially in transflection mode (e.g. when using ITO covered slides).

We thank the Reviewer for this comment and agree that considering approaches to correct for interferences, standing wave effects and scattering effects in future work will be important.

2. Reviewer #2

3.1 Comment 1

I regarded this manuscript as a methodology paper aiming at spatial lipidomics profiling at maximized analytical depth with QCL-MIR imaging-guided TIMS-MSI prm-PASEF analysis. After reading the revised manuscript and the detailed workflow, I kept my concerns about this technology. I would not recommend it for publication in Nature Communications.

Thanks for this honest statement. We are surprised though that all the additional examples, most importantly examination of spinal cord lesions in the EAE mouse model (new Figure 4 in the first revision) that we did especially in response to reviewer 2, are not mentioned at all. However, we are still convinced of the quality and novelty of our manuscript and aim to highlight this with the following answers.

3.2 Comment 2

The major concerns in the previous comments were NOT fully addressed (Comments #1, #2, #4, #5 and #6, and the major concern). I will not take the time to repeat these concerns.

We regret that the reviewer did not comment further on the detailed answers we provided for Comments #1, #2, #4, #5 and #6, and the major concern in the first review process. If we understand the meaning of “major concern” correctly, then this is the one we responded to in 2.1. to 2.4. But we are not sure.

To briefly summarize our response to reviewer 2, we added a “new biology” example (Fig. 4), several new Supp. Figures and a point-for-point response exceeding eight pages. Hence, we don’t know how to properly respond to “not fully addressed” without further hints.

Therefore, we still believe that we addressed his/her previous comments in the best possible way.

3.3 Comment 3: “Biological Replicates”

To illustrate with a simple example: the study includes enough technical replicates but lacks biological replicates. I highlighted this issue in my previous comments, yet it was entirely ignored by the authors. This disregard makes me feel that reviewing this work again is a waste of time and reflects a lack of respect, transparency and honest. Some other concerns were addressed in the similar way. I will not repeat again.

We are a bit surprised by this comment. First of all, we have carefully studied reviewer#2’s comments in the first revision again. However, the exact phrase “biological replicates” does not appear there once.

Nevertheless, we want to address it in detail: The data shown in the revised manuscript and also in the original manuscript were almost without exception based on biological replicates, mostly n=4. These are a few examples, where we specifically stated the use of biological replicates.

For the main manuscript:

- Line 356: ... “In two 60-week-old mouse kidneys we elucidated 101 and 112 sulfatides by TIMS-MSI in qTOF mode and 152 and 148 by QCL-MIR imaging-guided TIMS-MSI (**Table 1**).
- Line 856: ... “CCS values obtained from QCL-MIR imaging-guided MSI of four biological replicates were averaged” ...
- Line 876: ... “Two 12-week-old and two 60-week-old ARSA^{-/-}, two heterozygous (ARSA^{+/-}) and two wild-type (ARSA^{+/+}) kidney samples” ...
- Line 951: ... “12- or 60-week-old ARSA^{-/-} mice (n=2 each) were analyzed by LC-TIMS-MS or –MS² and MALDI-MSI (TIMS/qTOF).” (Table 1)

For the supplement:

- LC-TIMS-MS data analysis: ... “we identified 93–95 sulfatide species in two 60-week-old and 44-65 sulfatide species in two 12-week-old ARSA^{-/-} kidney samples.” ...
- Suppl. Fig. 3: ... “the procedure was repeated for four different 60-week-old mice, two wild-type (ARSA^{+/+}) mice and two arylsulfatase A-deficient (ARSA^{-/-}) mice.” ...

- Suppl. Fig. 8: ... “Comparison of glomeruli-containing kidney regions in wild-type (ARSA+/+; 60 weeks), ARSA+/- (12 weeks) and ARSA-/- mice (12 and 60 weeks) by TIMS-MSI (timsTOF fleX) and QCL-based MIR imaging microscopy” ...
- Suppl. Fig. 8: ... “**d**, Example data for tissue sections from three different mice (wild-type, heterozygotes, and knock-out).” ...
- Suppl. Fig. 13: ... “Sum intensity distribution of 87 sulfatides⁷ (internal standard normalized) obtained by MSI (timsTOF fleX, operated in qTOF mode) for ARSA-/- (60 weeks) vs. ARSA+/+ (60 weeks) and ARSA-/- (12 weeks) vs. ARSA+/- (12 weeks) mice (top).” ...
- Suppl. Fig. 15.
- Suppl. Fig. 18: ... “Ion images (timsTOF flex, qTOF mode) of m/z 892.619 (SM4 41:1;O3 [M-H]⁻) for two datasets of ARSA-/- (KO, left) and ARSA+/+ (WT, right).” ...

Moreover, in Suppl. Table 10 and 13 we present our data with uncertainties calculated based on $n=4$ biological replicates. The mass accuracy was also determined based on the mean m/z value across $n=4$ biological replicates. Presenting data with such statistical details is, according to our observation, highly uncommon in other literature in the field of mass spectrometry imaging.

Since statements for mouse kidneys (we examined 192 tissue sections in total!) and spheroids were obviously not labeled clearly enough, we revised the manuscript and Figure legends to emphasize the use of biological replicates more clearly. For example, we added remarks to biological replicates in line 176, 244, 270, 344, 521, 526, 552, 879, 975, and 984 for the main manuscript and all across the supplementary information. We hope that this helps avoid misunderstandings.

That said, we are happy to acknowledge that the EAE model case study that we added in revision as an example for how to address new biology discoveries, did lack biological replicates. We had only done technical replicates from the same mouse. We have now addressed this concern by upgrading our data to $n=3$ biological replicates (=different mice) for the EAE case study. In this context, we added an updated **Fig. 4** and added new **Supplementary Fig. 34-37** and new **Supplementary Table 14 and 20**.

2.4 Comment 4: “Additional Concerns”

In addition, there are additional concerns for the revised manuscript:

2.4.1 Concern 1

The authors mentioned that “Parallel reaction monitoring with parallel accumulation and serial fragmentation (prm-PASEF) of lipids/metabolites has so far only been outlined for TIMS-MS without spatial resolution.” In fact, similar spatially resolved analysis has already been conducted in previous work (<https://doi.org/10.1038/s41467-023-43298-9>). Therefore, the novelty of prm-PASEF for tissue-based MS/MS is diminished.

We thank the reviewer for this very important remark. It is embarrassing indeed, since I was a reviewer of the excellent Heuckeroth paper. Since I am an advocate of transparent peer review, my name even appears online next to this paper:

“Peer review information - Nature Communications thanks Carsten Hopf, Laura Sanchez and the other, anonymous, reviewer(s) for their contribution to the peer review of this work.”

I am not sure how this could happen. I thought that we had cited this break-through paper in TIMS-MSI. We regret this, and have we have updated the manuscript accordingly by adding a sentence in line 91ff.

2.4.2 Concern 2

MSI technology is developed for comprehensive spatial-omics profiling. Relying on ROIs defined by IRI for MSI-based spatial profiling would result in the loss of some potential biological information, making it insufficient to support new discoveries. The results shown in Supplementary Fig. 8 indicated that the MSI-unique regions were more extensive than the regions shared by both MSI and MRI. If the ROI was defined based on MRI results, the biological information in these MSI-unique regions would have been overlooked.

We thank the reviewer again for pointing this out. The entire topic of “omitting information by guidance” was extensively discussed in chapter 2.3 of the first revision.

At this point, we want to emphasize again that we never make a statement, that we don’t omit any information. Instead we repeatedly point out that there is a trade-off:

First, to our knowledge, this is the first demonstration of MIR-based detection of glomeruli in kidney. This is a feat in itself, and we hope that AI-based methods can boost the fidelity in the future. More importantly, as part of this trade-off, we accept that we may get some false-negatives for ROIs in QCL-MIR Imaging, e.g. 20% fewer glomeruli or EAE lesions. But it may not be very likely that these false negatives, i.e. structures not identified by QCL-MIR imaging, would contain biomarkers that the correctly identified glomeruli or lesions did not. In contrast, there is EVERY reason to believe that reinvesting the MSI measurement time saved by focusing on QCL-MIR imaging-defined ROIs into default on-tissue MS/MS by iprm-PASEF will effectively identify more biomarker candidates, because only MS/MS results in high confidence IDs.

Most importantly, we believe that the scientific community should be given the choice to judge for themselves.

Finally, we’d like to point out that already the MALDI process itself contains the same risk of potentiality overlooking or omitting information, since ionization is highly dependent on the selected chemical matrix etc. Depending on the selected matrix, it will likely happen that many molecules are not even ionized that would have been ionized with a different matrix chemical. This is a well-known trade-off that every mass spectrometrists has always accepted and dealt with. But nobody would question the MALDI MSI technique *per se* because of that.

2.4.3 Concern 3

A more detailed description of lipid identification is provided, but some issues remain. I just pointed out some examples. There are many others, and I will not take my time to repeat these concerns.

1) For lipids identified using prm-PASEF analysis in Supplementary Table 7, the authors choose M3 of PI 34:1 ([M-H]⁻). Why choose the M3 instead of M0?

2) PI 36:3 and PI 18:0/20:3 were NOT isomeric structures. In fact, their mass differs by 28 Da. However, in Supplementary Table 8, they were annotated in a chimeric spectrum.

3) The authors set the isolation window as ± 1.1 Da. Why did the MS/MS spectrum annotated as PI 34:1 showed interference from PI 40:3 fragments?

We kindly thank the Reviewer for also thoroughly reviewing the Supplement.

- 1) To avoid omitting any information, we selected the precursors for iprm-PASEF analysis strictly according to the results of the LASSO regression (see **Supplementary Fig. 6d and 6f**) This means that occasionally features that correspond to isotopes of molecules could turn out to reveal the highest feature importance scores. To address this issue in the future, implementing an isotopic filtering algorithm would be an option. However, this would also introduce a potential loss of information because overlapping features might then also be excluded. For example, in Suppl. Table 9, we show that M2 of PI 40:4 overlaps with M0 of PI 40:3. Isotopic filtering would have prevented this finding. We added a sentence in the caption of **Supplementary Fig. 6** to make this clear.
- 2) We corrected this typing error in the revised manuscript. It is indeed PI 16:0/20:3 instead of PI 18:0/20:3.
- 3) We corrected this typing error as well. It is PI 40:4 instead of PI 34:1. We updated the caption accordingly and added the predominant isomeric structures.

2.4.4 Concern 4

When comparing lipid changes between EAE mice and control mice, individual differences were not considered, and no biological replicates were included. It would be better to characterize the lipid remodeling in spinal cord with the lesions and their adjacent normal areas from the same mouse.

Regarding biological replicates (see above), we updated Fig. 4 and added new Supplementary Fig. 34 to demonstrate that our findings are based on n=3 biological replicates. The data presented in new Supplementary Fig. 36 and 37 is obtained from n=3 biological replicates. This also includes a recalculation of the Volcano plot. To this end we added also a description of the volcano analysis in the Supplement.

Whether or not a lesion should be compared with tissue from a WT mouse or “normal areas” from the same mouse, in our opinion, is a matter of debate. We cannot assume that apparently “normal” tissue adjacent to a lesion, even if it appears normal by conventional histology, is really normal. There may be merit in both approaches. Therefore, we did both and added two New Suppl. **Figures, 36 (lesion versus healthy control) and 37 (lesion versus adjacent tissue)**. In fact, results were very similar.

2.5 Comment 5

Additionally, in Line 360, the authors suggested that lipids detected only in TIMS-MSI were due to the different mass and mobility ranges in LC-TIMS-MS. It is better to use the same mass and mobility ranges in LC-TIMS-MS analysis for comparison. With the help of LC separation, it could have assisted in confirming whether SM1a/b originated from the in-source fragmentation of SB1a.

We agree with the reviewer that only the same mobility and m/z ranges should be used to compare LC-TIMS-MS and TIMS-MSI. The comparison we show in **Fig. 2g** is based entirely on comparable settings. We have revised the manuscript to make this unambiguously clear.

In addition, we have now revised **Supplementary Fig. 24**.

The additional analyses in relation to SM1a/b were carried out in the course of the first revision with the ambition of achieving maximum coverage. We are aware that no LC-TIMS-MS data is available for this. Therefore, we are willing to remove this data upon editorial request. However, we want to point out that this data is still highly valuable to further improve state-of-the-art CCS prediction tools, independent of whether the sulfatides originate from in-source-decay or not and therefore important for the community.

3.6 Final Comment

Overall, I kept my concerns about the novelty and essentiality of the developed method. I am skeptical about the accuracy of the lipid identification. The validations of quantitative accuracy and reproducibility are needed. The application study lacks of biological replication. And many others!

We thank the Reviewer for stating his/her concerns regarding the lipid identification, even though we don't share them.

Our lipid identifications are based on the three pillars. First, we achieved mass accuracies of sub 3 ppm at the MS¹ level for almost every identified lipid. Second, we acquired iprm-PASEF-derived MS² spectra and presented the elucidation of the fragment patterns either in now **Suppl. Figs. 19-23, 25**, and new **35** or **Suppl. Tables 2-9** and **15-21**. For each sulfatide species we acquired MS² spectra for at least n=2 biological replicates. We added **new Supplementary Fig. 17** to show the comparability of our "on-tissue" MS² analysis to the analysis of a standard compound purchased from Avanti Polar Lipids. We also added **new Supplementary Fig. 18** to demonstrate the reproducibility of the MS² spectra across different animal individuals (biological replicates).

As the third pillar, we acquired ion mobility data for n=4 biological replicates and calculated the mean CCS values for the respective sulfatides. In **Fig. 3bc** we show the validation of these CCS data by comparison against the most accurate CCS prediction tool (AllCCS2) and more importantly against LC-TIMS-MS data acquired in the Bindila lab, a leading lipidomics lab (Lerner et al., Four-dimensional trapped ion mobility spectrometry lipidomics for high throughput clinical profiling of human blood samples. Nat Commun. 2023 Feb 20;14(1):937. doi: 10.1038/s41467-023-36520-1). The mean relative errors of these comparisons are 0.52% and 1.38% respectively, which highlights our accuracy in lipid identification.

At this point, we want to mention that we already showed in the original manuscript in **Fig. 2** a detailed evaluation of reproducibility across n=4 biological replicates. Everything in regard to biological replicates was already discussed in Comment 3.

Point-for-Point Reply to Reviewers – 3rd Revision

Deep MALDI-MS Spatial ‘Omics guided by Quantum Cascade Laser Mid-infrared Imaging Microscopy

Contents

1. Reviewer #4	1
1.1 Comment	1
2.2 Major Comment 1.....	2
2.3 Major Comment 2.....	2
2.4 Major Comment 3.....	2
2.5 Minor Comments.....	3
2.6 Further Comments.....	3
2.7 Comments regarding other reviewer’s comments.....	4

3rd Revision

1. Reviewer #4

1.1 Comment

The manuscript by Gruber et al details the introduction and application of quantum cascade laser mid-infrared imaging microscopy (QCL-MIR imaging). The authors developed QCL-MIR imaging-guided MSI workflows and computational tools for spatially resolved deep lipidomics profiling and made them and extensive spatial lipidomics data available as a community resource. QCL-MIR imaging-guided spatially resolved on-tissue MS2 of lipids by iprm- PASEF using ion mobilograms was performed and benched marked using a ground truth animal model where specific lipids, sulfatides, were known to be dysregulated and previously characterized. Overall this is a very well done study I have tried to parse out my own critiques and comments with also being asked to evaluate prior reviewer comments without being an original reviewer.

We sincerely thank the reviewer for his/her thoughtful and thorough review of our manuscript. We appreciate his/her accurate summary highlighting the development of QCL-MIR imaging for guidance of associated MSI/iprm-PASEF workflows, the introduction computational tools made available as a community resource, and the benchmarking using the ground truth animal model for spatially resolved lipidomics. His/her assessment of the study as "very well done" is very encouraging.

2.2 Major Comment 1

While the authors do address adding the 2023 Nat. Commun SIMSEF paper, a statement remains in the abstract that should be altered to match the updated introduction, that current MSI only offers “putative” molecular identifications” is incorrect a number of instruments are capable of acquiring MS/MS and ion mobility CCS values. This sentence should be adjusted in light of the SIMSEF paper that is later mentioned and discussed. I do agree that most people still don’t use this but I believe this is more of a data size and handling issue rather than not being possible on a number of different instruments.

We have deleted “and often delivers only “putative” molecular identifications” in the abstract to account for this valid concern.

We completely agree with the reviewer that while methods like SIMSEF for MS/MS and ion mobility exist because of time requirements/speed and data size they are not commonly used. Therefore, we had written in the introduction: “Therefore, in practice, most MALDI-MSI studies with high-performance instruments today appear to use slow FTMS imaging or timsTOF-MSI in TIMS-off/qTOF mode without using collisional cross section (CCS) information. Because of time constraints and despite the lack of HPLC separation in MSI, analytical capabilities for spatially resolved accurate mass determination (in FTMS imaging), on-tissue fragmentation analysis, and ion mobility separation of isobaric compounds (both in TIMS-MSI) are often not used.”

This is precisely why we argue that a QCL-MIR prescan can help to alleviate the time/data size issue.

2.3 Major Comment 2

“Dedicated computational tools for information-rich and high-throughput QCL-MIR imaging-guided MSI are lacking” The experimental data and the code should be made publicly available reported in this manuscript. A major promise and premise of this workflow and the novelty lies in the integration of the data as outlined in SI Figure 1. It is unclear how exactly the data analysis proceeds and the timeline for the analysis. As it stands, it is unclear exactly how one would replicate these results from the IT workflow standpoint, which is where the major novelty of this manuscript lies. Please make this and the data publicly available. This would seem like a major impediment to wide adoption of this technology to not readily release the code needed.

We had planned to make code and data available upon acceptance of the manuscript all along. Here, we have updated the data and code availability statements accordingly. To further support transparency and usability, we've added a graphical overview and incorporated the requested details into Supplementary Fig. 1b. Additionally, our GitHub repository includes a brief hands-on guide to help users get started with the underlying software. Upon publication of the manuscript, the GitHub repository will be made publicly available to support open access and hopefully wide adoption of this code/workflow.

2.4 Major Comment 3

In SI Figure 1, do parts 3 and 4 segmentation and ROI transfer occur offline from both instruments? Can the authors comment on the time for each of these steps to give a feel for how long this might take and storage recommendations for the tissues during the offline analysis?

Yes, the steps 3 and 4 occur offline from both instruments since up-to-now, there has been no software solution for this purpose. We have added the approximate time requirements for each step in Supplementary Fig. 1b. However, we would like to point out that the time steps, in particular signal processing, is case-dependent. In addition, we have added a statement regarding the storage recommendations during the offline analysis in the method section:

“To prevent tissue degradation due to environmental conditions, the slides were either processed immediately or carefully placed in a slide mailer, vacuum-sealed and stored at -80°C for optimal sample preservation.”

2.5 Minor Comments

2.5.1 Minor Comment 1

Regarding SI Figure 5, could the authors comment on the error associated with the QCL-MIR measurements. These differences they highlight seem very minor or within the margin of error in other types of measurements. I am not expressly familiar with this measurement so providing some level of acceptable interpretation and error would be helpful for a generalist reader.

The significance and associated uncertainty of the QCL-MIR measurements presented in Suppl. Fig. 5b are addressed and discussed in Suppl. Fig. 13c and 13d. The examination of the (semi-)quantitative data evaluation provides a key and exemplary analysis within the study. To highlight this, we have updated the figure caption of Suppl. Fig. 5b.

2.5.2 Minor Comment 2

On a couple of occasions the authors refer to “computationally dilated” do they mean enlarged, the meaning of this is unclear.

Yes, we have re-phrased it throughout the manuscript and used the wording *expanded*.

2.6 Further Comments

2.6.1 Comment 1

I appreciated the including of data in SI Figures 3-4 this was nicely done and highlights that the QCL-MIR is non destructive to the sample and the lipids being queried.

We thank the reviewer for valuing the work we put in this particular step.

2.6.2 Comment 2

The amount of data provided in the SI provides a high level of rigor to this manuscript. The inclusion of the different controls with replicates and the MS/MS spectra is well done including key experimental details such as the CE.

We thank the reviewer for highlighting our detailed labelling.

2.6.3 Comment 3

The data comparisons and increased identifications presented in the section entitled “QCL-MIR imaging-guided trapped TIMS-MSI enables extensive fragmentation analysis on tissue and sulfatide identifications on par with LC-based 4D-TIMS-PASEF” highlights the utility of this method, although it was also nice to see that the identification numbers weren’t vastly different but enough to highlight that targeted methods for directed analysis are beneficial especially for a more comprehensive analysis than is currently being conducted during most general experiments. This strikes a nice balance for utility and need.

We thank the reviewer for this comment.

2.7 Comments regarding other reviewer’s comments

2.7.1 Comment 1

As a new reviewer of this manuscript, regarding 2.4.2 Concern 2, the authors do an excellent job of highlighting the tradeoffs between a fully untargeted 4D MALDI-TIMS-qTOF experiment and time and using structural features provided by MIR to guide a targeted in-depth analysis. At no point do they propose that this methodology should replace what labs are currently doing, but rather this new methodological workflow provides a different, orthogonal route for probing biomedically relevant questions.

We thank the reviewer for pointing this out.

2.7.2 Comment 2

The lipid identifications are well done and sufficient data including MS/MS and CCS values in relevant models in which these molecules are known to be implicated is provided. I have no concerns regarding the assignment of the lipid species in this manuscript. The SI was expensive and comprehensive.

We highly appreciate the recognition of the extensive data in the SI and the methodological rigor. We would also like to thank the reviewer in particular for his/her supportive words regarding the comments from the previous round and his/her agreement with our positioning of the methodological workflow.